# Decentralized Transformers with Centralized Aggregation are Sample-Efficient Multi-Agent World Models

## Abstract

Learning a world model for model-free Reinforcement Learning (RL) agents can significantly improve the sample efficiency by learning policies in imagination. However, building a world model for Multi-Agent RL (MARL) can be particularly challenging due to the scalability issue in a centralized architecture arising from a large number of agents, and also the non-stationarity issue in a decentralized architecture stemming from the inter-dependency among agents. To address both challenges, we propose a novel world model for MARL that learns decentralized local dynamics for scalability, combined with a centralized representation aggregation from all agents. We cast the dynamics learning as an auto-regressive sequence modeling problem over discrete tokens by leveraging the expressive Transformer architecture, in order to model complex local dynamics across different agents and provide accurate and consistent long-term imaginations. As the first pioneering Transformer-based world model for multi-agent systems, we introduce a Perceiver Transformer as an effective solution to enable centralized representation aggregation within this context. Main results on Starcraft Multi-Agent Challenge (SMAC) and additional results on MAMujoco show that it outperforms strong model-free approaches and existing model-based methods in both sample efficiency and overall performance.

## 1 Introduction

Multi-Agent Reinforcement Learning (MARL) has made remarkable progress, which was driven largely by model-free algorithms (Nguyen et al., 2020). However, due to the complexity of multi-agent systems arising from large state-action space and partial observability, such algorithms usually demand extensive interactions to learn coordinative behaviors (Hernandez-Leal et al., 2020). A promising solution is building a world model that approximates the environment, which has exhibited its superior sample efficiency compared to model-free approaches in single-agent RL (Hafner et al., 2020; Łukasz Kaiser et al., 2020; Hafner et al., 2021; 2023; Hansen et al., 2022; 2024). However, extending the design of world model in single-agent domain to the multi-agent context encounters significant challenges due to the unique biases and characteristics inherent to multi-agent environments.

The challenges primarily stem from two different means for multi-agent dynamics learning: *centralized* and *decentralized*. Learning a world model to approximate the *centralized* dynamics encapsulates the inter-dependency between agents but struggles to be scalable to an increasing number of agents, which leads to the exponential surge in spatial complexity (Hernandez-Leal et al., 2020; Nguyen et al., 2020). Conversely, applying a *decentralized* world model to approximating the local dynamics of each agent mitigates the scalability issue yet incurs non-stationarity, as unexpected interventions from other agents may occur in each agent's individual environment (Oliehoek et al., 2016). Furthermore, beyond these unique challenges inherent in modeling multi-agent dynamics, existing model-based MARL approaches (Willemsen et al., 2021; Egorov & Shpilman, 2022; Xu et al., 2022) excessively neglect the fact that the policy learned in imaginations of the world model heavily relies on the quality of imagined trajectories (Micheli et al., 2023). It thereby necessitates accurate long-term prediction, especially with respect to the non-stationary local dynamics. Inspired by the capability of Transformer (Vaswani et al., 2017) in modeling complex discrete sequences and long-term dependency (Brown et al., 2020; Devlin et al., 2019; Micheli et al., 2023), we seek to construct a

Transformer-based world model within the multi-agent context for *decentralized* local dynamics together with *centralized* feature aggregation, combining the benefits of two distinctive designs.

In this paper, we introduce MARIE (Multi-Agent auto-Regressive Imagination for Efficient learning), the first Transformer-based multi-agent world model for sample-efficient policy learning. Specifically, **the highlights of this paper are:**

1. To tackle the inherent challenges within the multi-agent context, we build an effective world model via scalable *decentralized* dynamics modeling and essential *centralized* representation aggregating, which mirrors the principle of Centralized Training and Decentralized Execution.

2. To enable accurate and consistent long-term imaginations from the non-stationary local dynamics, we cast the *decentralized* dynamics learning as sequence modeling over discrete tokens by leveraging highly expressive Transformer architecture as the backbone. In particular, we successfully present the first Transformer-based world model for multi-agent systems.

3. While it remains open for how to effectively enable *centralized* representation with the Transformer as the backbone, we achieve it by innovatively introducing a Perceiver Transformer (Jaegle et al., 2021) for efficient global information aggregation across all agents.

4. Experiments on the Starcraft Multi-Agent Challenge (SMAC) benchmark in low data regime and additional experiments on MAMujoco show MARIE outperforms both model-free and existing model-based MARL methods w.r.t. both sample efficiency and overall performance and demonstrate the effectiveness of MARIE.

## 2 RELATED WORKS AND PRELIMINARIES

**Multi-Agent Reinforcement Learning.** In a model-free setting, a typical approach for cooperative MARL is centralized training and decentralized execution (CTDE), which tackles the scalability and non-stationarity issues in MARL. During the training phase, it leverages global information to facilitate agents' policy learning; while during the execution phase, it blinds itself and has only access to the partial observation around each agent for multi-agent decision-making. Model-free MARL methods with this paradigm can be divided into 2 categories: value-based (Sunehag et al., 2018; Rashid et al., 2018; Son et al., 2019; Wang et al., 2021) and policy-based (Lowe et al., 2017; Foerster et al., 2018; Iqbal & Sha, 2019; Ryu et al., 2020; Liu et al., 2020; Kuba et al., 2021; Peng et al., 2021; Yu et al., 2022; Zhang et al., 2024b;a). In contrast to model-free approaches, model-based MARL algorithms remain fairly understudied. MAMBPO (Willemsen et al., 2021) incorporates MBPO-style (Janner et al., 2019) techniques into multi-agent policy learning under the CTDE framework. Tesseract (Mahajan et al., 2021) introduces the tensorised Bellman equation and evaluates the Q-value function using Dynamic Programming (DP) together with an estimated environment model. Similar to our setting where agents learn inside of an approximate world model, MAMBA (Egorov & Shpilman, 2022) integrates the backbone proposed in DreamerV2 (Hafner et al., 2021) with an attention mechanism across agents to sustain an effective world model in environments with an arbitrary number of agents, which leads to notably superior sample efficiency to existing model-free approaches. In terms of model-based algorithm coupled with planning, MAZero (Liu et al., 2024) expands the MCTS planning-based Muzero (Schrittwieser et al., 2020) framework to the model-based MARL settings. However, learning-based or planning-based policies in these two approaches are both overly coupled with their world models, downgrading their inference efficiency and further limiting expansion in combinations with other popular model-free approaches. To the best of our knowledge, we are the first to expand the Transformer backbone-based world model within the multi-agent context.

**Learning behaviors within the imagination of world models.** The Dyna architecture (Sutton, 1991) first emphasizes the utility of an estimated dynamics model in facilitating the training of the value function and policy. Inspired by the cognitive system of human beings, the concept of world model (Ha & Schmidhuber, 2018) is initially introduced by composing a variational Auto-Encoder (VAE) (Kingma & Welling, 2014) and a recurrent network to mimic the complete environmental dynamics, then an artificial agent is trained entirely inside the hallucinated imagination generated by the world model. SimPLe (Łukasz Kaiser et al., 2020) shows that a PPO policy (Schulman et al., 2017) learned

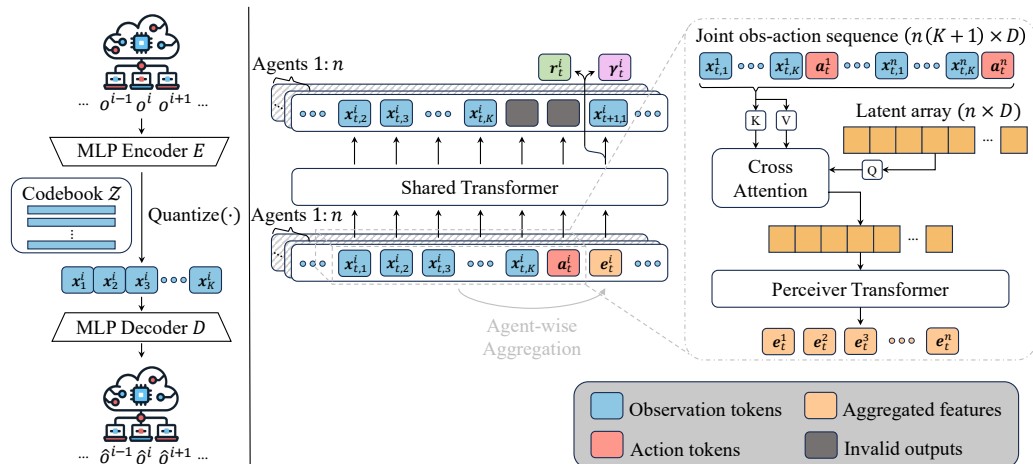

Figure 1: Overview of the proposed world model architecture in MARIE. VQ-VAE (**left**) maps local observations $o^i$ of each agent $i$ into discrete latent codes $(x_1^i, ..., x_K^i)$, where $(E, D, \mathcal{Z})$ is shared across all agents. Together with discrete actions, this process forms local discrete sequences $(..., x_{t,1}^i, ..., x_{t,K}^i, a_t^i, ...)$ of each agent. Then the Perceiver (**right**) performs aggregation of joint discrete sequences of all agents $(x_{t,1}^1, ..., x_{t,K}^1, a_t^1, ..., x_{t,1}^n, ..., x_{t,K}^n, a_t^n)$ independently at each timestep $t$, and inserts the aggregated global representations $(e_t^1, e_t^2, ..., e_t^n)$ into original local discrete sequences respectively. The resulting sequences $(..., x_{t,1}^i, ..., x_{t,K}^i, a_t^i, e_t^i...)$ contain rich information between transitions in local dynamics and are fed into the shared Transformer (**middle**), which learns observation token predictions in an autoregressive manner. Predictions of individual reward $r_t^i$ and discount $\gamma_t^i$ at timestep $t$ are computed based on all historical sequence $(x_{\leq t,1}^i, ..., x_{\leq t,K}^i, a_{\leq t}^i, e_{\leq t}^i)$.

in a predictive model deliverer a super-human performance in Atari domains. Dreamer (Hafner et al., 2020) builts the world model upon a Recurrent State Space Model (RSSM) (Hafner et al., 2019) that combines the deterministic latent state with the stochastic latent state to allow the model to not only capture multiple futures but also remember information over multi-steps. DreamerV2 (Hafner et al., 2021) further demonstrates the advantage of discrete latent states over Gaussian states. For MARL, MAMBA (Egorov & Shpilman, 2022) extends DreamerV2 to multi-agent contexts by using RSSM, underscoring the potential of multi-agent learning in the imagination of world models. Recently, motivated by the success of the Transformer (Vaswani et al., 2017), TransDreamer (Chen et al., 2022) and TWM (Robine et al., 2023) explored variants of DreamerV2, wherein the backbones of the world model were substituted with Transformers. Instead of incorporating deterministic and stochastic latent states, IRIS (Micheli et al., 2023) applies the Transformer to directly modeling sequences of observation tokens and actions of single-agent RL and achieves impressive results on Atari-100k. In contrast, the proposed MARIE concentrates on establishing effective Transformer-based world models in multi-agent contexts with shared dynamics and global representations.

**Preliminaries.** We focus on fully cooperative multi-agent systems where all agents share a team reward signal. We formulate the system as a decentralized partially observable Markov decision process (Dec-POMDP) (Oliehoek et al., 2016), which can be described by a tuple $(\mathcal{N}, \mathcal{S}, \mathcal{A}, P, R, \mathbf{\Omega}, \mathcal{O}, \gamma)$. $\mathcal{N} = \{1, ..., n\}$ denotes a set of agents, $\mathcal{S}$ is the finite global state space, $\mathcal{A} = \prod_{i=1}^n \mathcal{A}^i$ is the product of finite actions spaces of all agents, i.e., the joint action space, $P : \mathcal{S} \times \mathcal{A} \times \mathcal{S} \to [0, 1]$ is the global transition probability function, $R : \mathcal{S} \times \mathcal{A} \to \mathbb{R}$ is the shared reward function, $\mathbf{\Omega} = \prod_{i=1}^n \Omega^i$ is the product of finite observation spaces of all agents, i.e., the joint observation space, $\mathcal{O} = \{\mathcal{O}^i, i \in \mathcal{N}\}$ is the set of observing functions of all agents. $\mathcal{O}^i : \mathcal{S} \to \Omega^i$ maps global states to the observations for agent $i$, and $\gamma$ is the discount factor. Given a global state $s_t$ at timestep $t$, agent $i$ is restricted to obtaining solely its local observation $o_t^i = \mathcal{O}^i(s_t)$, takes an action $a_t^i$ drawn from its policy $\pi^i(\cdot|o_{\leq t}^i)$ based on the history of its local observations $o_{\leq t}^i$, which together with other agents' actions gives a joint action $\boldsymbol{a}_t = (a_t^1, ..., a_t^n) \in \mathcal{A}$, equivalently drawn from a joint policy $\boldsymbol{\pi}(\cdot|\boldsymbol{o}_{\leq t}) = \prod_{i=1}^n \pi^i(\cdot|o_{\leq t}^i)$. Then the agents receive a shared reward $r_t = R(s_t, \boldsymbol{a}_t)$, and the environment moves to next state $s_{t+1}$ with probability $P(s_{t+1}|s_t, \boldsymbol{a}_t)$. The aim of all agents is to learn a joint policy $\boldsymbol{\pi}$ that maximizes the expected discounted return $J(\boldsymbol{\pi}) = \mathbb{E}_{s_0, \boldsymbol{a}_0, ... \sim \boldsymbol{\pi}} [\sum_{t=0}^\infty \gamma^t R(s_t, \boldsymbol{a}_t)]$.

## 3 METHODOLOGIES

Our approach comprises three typical parts: (i) collecting experience by executing the policy, (ii) learning the world model from the collected experience, and (iii) learning the policy via imagination inside the world model. Throughout the process, the historical experiences stored in the replay buffer are used for training the world model only, while policies are learned from unlimited imagined trajectories from the world model. In the following, we first describe three core components of our world model in §3.1 and §3.2, and give an overview of the proposed world model in Fig. 1. Then we describe the policy-learning process inside the world model in §3.3. The comprehensive details of the model architecture and hyperparameter are provided in §A.

### 3.1 DISCRETIZING OBSERVATION

We consider a trajectory $\tau^i$ of agent $i$ consists of $T$ local observations and actions, as

$$\tau^i = (o_1^i, a_1^i, \ldots, o_t^i, a_t^i, \ldots, o_T^i, a_T^i).$$

To utilize the expressive Transformer architecture, we need to express the trajectory into a discrete token sequence for modeling. Accounting for continuous observations, a prevalent but naive approach for discretization involves discretizing the scalar into one of $m$ fixed-width bins in each dimension independently (Janner et al., 2021). However, when faced with a higher dimension of the observation, such discretization would encode the observation with more tokens, leading to higher computational complexity of the later sequence modeling via the Transformer, which necessitates an approach that uses a discrete codebook of learned compact representations. To this end, we employ the idea from neural discrete representation learning (van den Oord et al., 2017), and learn a Vector Quantised-Variational AutoEncoder (VQ-VAE) to play a role that resembles the tokenizer in Natural Language Processing (Devlin et al., 2019; Brown et al., 2020). The VQ-VAE is composed of an encoder $E$, a decoder $D$, and a codebook $\mathcal{Z}$. We define the discrete codebook $\mathcal{Z} = \{z_j\}_{j=1}^N \subset \mathbb{R}^{n_z}$, where $N$ is the size of the codebook and $n_z$ is the dimension of codes. The encoder $E$ takes an observation $o^i \in \mathbb{R}^{n_{\text{obs}}}$ as input and outputs a $K$ $n_z$-dimensional latents $\hat{z}^i \in \mathbb{R}^{K \times n_z}$ reshaped from the direct outputs of encoder. Subsequently, the tokens $\{x_k^i\}_{k=1}^K \in \{0, 1, ..., N-1\}^K$ for representing $o^i$ is obtained by a nearest neighbour look-up using the codebook $\mathcal{Z}$ where $x_k^i = \arg\min_j \|\hat{z}_k^i - z_j\|$. Then the decoder $D : \{0, 1, ..., N-1\}^K \to \mathbb{R}^{n_{\text{obs}}}$ converts $K$ tokens back into an reconstructed observation $\hat{o}^i$. By learning this discrete codebook, we compress the redundant information via a succinct sequence of tokens, which helps improve sequence modeling. See §4.2 for a discussion.

### 3.2 MODELING LOCAL DYNAMICS WITH GLOBAL REPRESENTATIONS

Here, we consider discrete actions like those in SMAC, and the continuous actions can also be discretized by splitting the value in each dimension into fixed bins (Janner et al., 2021; Brohan et al., 2023). Therefore, a trajectory $\tau^i$ of agent $i$ can be treated as a sequence of tokens,

$$\tau^i = (\ldots, o_t^i, a_t^i, \ldots) = (\ldots, x_{t,1}^i, x_{t,2}^i, \ldots, x_{t,K}^i, a_t^i, \ldots) \tag{1}$$

where $x_{t,j}^i$ is the $j$-th token of the observation of agent $i$ at timestep $t$. Given arbitrary sequences of observation and action tokens in Eq. (1), we try to learn over discrete multimodal tokens.

The world model consists of a tokenizer to discrete the local observation, a Transformer to learn the local dynamics, an agent-wise representation aggregation module, and predictors for the reward and discount. The Transformer $\phi$ predicts the future local observation $\{\hat{x}_{t+1,j}^i\}_{j=1}^K$, the future individual reward $\hat{r}_t^i$ and discount $\hat{\gamma}_t^i$, based on the agent's individual historical observation-action history $(x_{\leq t,\cdot}^i, a_{\leq t}^i)$ and aggregated global feature $e_t^i$ of the agent. The modules are shown in Eqs. (2)–(5).

Transition: $\quad \hat{x}_{t+1,\cdot}^i \sim p_\phi(\hat{x}_{t+1,\cdot}^i | x_{\leq t,\cdot}^i, a_{\leq t}^i, e_{\leq t}^i)$ with $\hat{x}_{t+1,k}^i \sim p_\phi(\hat{x}_{t+1,k}^i | x_{\leq t,\cdot}^i, a_{\leq t}^i, e_{\leq t}^i, x_{t+1,<k}^i)$

$$\tag{2}$$

Reward: $\quad \hat{r}_t^i \sim p_\phi(\hat{r}_t^i | x_{\leq t,\cdot}^i, a_{\leq t}^i, e_{\leq t}^i) \tag{3}$

Discount: $\quad \hat{\gamma}_t^i \sim p_\phi(\hat{\gamma}_t^i | x_{\leq t,\cdot}^i, a_{\leq t}^i, e_{\leq t}^i) \tag{4}$

Aggregation: $\quad (e_t^1, e_t^2, ..., e_t^n) = f_\theta(x_{t,1}^1, x_{t,2}^1, ..., x_{t,K}^1, a_t^1, ..., x_{t,1}^n, x_{t,2}^n, ..., x_{t,K}^n, a_t^n) \tag{5}$

**Transition Prediction.** In the transition prediction in Eq. (2), the $k$-th observation token is additionally conditioned on the tokens that were already predicted $x_{t+1,<k}^i \triangleq (x_{t+1,1}^i, x_{t+1,2}^i, ..., x_{t+1,k-1}^i)$, ensuring the autoregressive token prediction to facilitate modeling over the trajectory sequence. Inter-step auto regression is as intuitive as predicting the future based on all information in the past while intra-step auto regression can be interpreted as learning how to compose the language provided by VQ-VAE to correctly express the observation within a certain timestep, since the tokens for encoding observations can be viewed as a special inner language like the human's.

**Discount and Reward Prediction.** The discount predictor outputs a Bernoulli likelihood and lets us estimate the probability of an individual agent's episode ending when learning behaviors from model predictions. And we simply adopt a smooth L1 loss for training the prediction of reward.

**Agent-wise Aggregation.** Due to the partial environment, the non-stationarity issue stems from the sophisticated agent-wise inter-dependency on local observations generation. To address it, we introduce a Perceiver (Jaegle et al., 2021) to perform agent-wise representation aggregation which plays a similar role to communication. To sustain the decentralized manner in transition prediction, we hope every agent can possess its own inner perception of the whole situation. Nonetheless, with discrete representation for local observation, the observation-action pair of agent $i$ at timestep $t$ is projected into a sequence $(x_{t,1}^i, x_{t,2}^i, ..., x_{t,K}^i, a_t^i)$ of length $K+1$. It leads to a joint observation-action sequence of length $n(K+1)$ at a timestep, which linearly scales with the number of agents.

A naive approach for extracting aggregated feature for each agent is using self-attention (Egorov & Shpilman, 2022; Liu et al., 2024) which takes as input this sequence of length $n(K+1)$ and outputs a sequence of the same length containing aggregated features of all agents, described as

$$(x_{t,1}^1, ..., x_{t,K}^1, a_t^1, ..., x_{t,1}^n, ..., x_{t,K}^n, a_t^n) \xrightarrow[\text{Aggregating}]{\text{Self-Attention}} (e_{t,1}^1, ..., e_{t,K}^1, e_{t,K+1}^1, ..., e_{t,1}^n, ..., e_{t,K}^n, e_{t,K+1}^n).$$

where $e_{t,j}^i$ is the $j$-th aggregated feature for agent $i$ at timestep $t$. However, when composing the informative sequence of local trajectories by insert these aggregated features into the sequence of length $H(K+1)$ in Eq. (1), the length of local sequence involving aggregated features would be twice longer, i.e., $2H(K+1)$. Due to the quadratic computational complexity of Transformer, it may hinder the efficient sequence modeling over this sequence.

To this end, we choose the Perceiver as the agent-wise representation aggregation module, which excels at dealing with the case that the size of inputs scales linearly and then generates a compact output sequence. Equipped with a flexible querying mechanism and self-attention mechanism, the Perceiver aggregates the joint representation sequence $(x_{t,1}^1, x_{t,2}^1, ..., x_{t,K}^1, a_t^1, ..., x_{t,1}^n, x_{t,2}^n, ..., x_{t,K}^n, a_t^n)$ of length $n(K+1)$ into a sequence of $n$ features $(e_t^1, e_t^2, ..., e_t^n)$,

$$(x_{t,1}^1, ..., x_{t,K}^1, a_t^1, ..., x_{t,1}^n, ..., x_{t,K}^n, a_t^n) \xrightarrow[\text{Aggregating}]{\text{Perceiver}} (e_t^1, e_t^2, ..., e_t^n)$$

where each feature $e_t^i$ serves as an intrinsic global abstraction of the environmental contexts perceived from agent $i$'s viewpoint. By introducing Perceiver, we provide a feasible solution for reducing the modeling complexity when using transformer-based local dynamics.

**Overall Learning Objective.** The world model $\phi$ is trained with trajectory segments of a fixed horizon $H$ sampled from the replay buffer $\mathcal{D}$ in a self-supervised manner. The transition predictor, discount predictor, and reward predictor are optimized to maximize the log-likelihood of their corresponding targets:

$$\mathcal{L}_{\text{Dyn}}(\phi, \theta) = \mathbb{E}_{i \sim \mathcal{N}} \mathbb{E}_{\tau^i \sim \mathcal{D}} \Big[ \sum_{t=1}^{H} \underbrace{- \log p_\phi(r_t^i | x_{\leq t,\cdot}^i, a_{\leq t}^i, e_t^i)}_{\text{reward loss}} \underbrace{- \log p_\phi(\gamma_t^i | x_{\leq t,\cdot}^i, a_{\leq t}^i, e_t^i)}_{\text{discount loss}}$$

$$\underbrace{- \Big( \sum_{k=1}^{K} \log p_\phi(x_{t+1,k}^i | x_{\leq t,\cdot}^i, a_{\leq t}^i, e_t^i, x_{t+1,<k}^i) \Big)}_{\text{transition loss}} \Big] \quad (6)$$

$$\text{where } (e_t^1, e_t^2, ..., e_t^n) = f_\theta(x_{t,1}^1, x_{t,2}^1, ..., x_{t,K}^1, a_t^1, ..., x_{t,1}^n, x_{t,2}^n, ..., x_{t,K}^n, a_t^n), \forall t.$$

We jointly minimize this loss function in Eq. (6) with respect to the model parameters of local dynamics (i.e., $\phi$) and global representation (i.e., $\theta$) using the Adam optimizer (Kingma & Ba, 2015).

Figure 2: Imagination procedure in MARIE. We unroll the imagination of all agents $\{1, ..., n\}$ in parallel. Initially, each agent's observation is derived from a joint observation sampled from a replay buffer. A policy, depicted in red arrows, generates actions based on reconstructed observations. Then, the Perceiver integrates joint actions and observations into global representations from each agent, appending them to each agent's local sequence. The Transformer then predicts individual rewards and discounts, depicted by green and purple arrows respectively, while generating next observation tokens for each agent in an autoregressive manner, shown by blue arrows. This parallel imagination iterates for $H$ steps. The policies $\pi_\psi^{1:n}$ are exclusively trained using imagined trajectories.

### 3.3 LEARNING BEHAVIOURS IN IMAGINATION

We utilize the Actor-Critic framework to learn the behavior of each agent, where the actor and critic are parameterized by $\psi$ and $\xi$, respectively. In the following, we take agent $i$ as an exemplar case for clarity and omit the superscript for denoting the index of the agent to avoid potential confusion. Benefited from the shared local dynamics, the local trajectories of all agents are imagined in parallel, as illustrated in Fig. 2. At timestep $t$, the actor takes a reconstructed observation $\hat{o}_t$ as input, and samples an action $a_t \sim \pi_\psi(a_t|\hat{o}_t)$. The world model then predicts the individual reward $\hat{r}_t$, individual discount $\hat{\gamma}_t$ and next local observation $\hat{o}_{t+1}$. Starting from initial observations sampled from the replay buffer, this imagination procedure is rolled out for $H$ steps. To stimulate long-horizon behavior learning, the critic accounts for rewards beyond the fixed imagination horizon and estimates the individual expected return $V_\xi(\hat{o}_t) \simeq \mathbb{E}_{\pi_\psi}[\sum_{l \geq t} \gamma^{l-t} \hat{r}_l]$.

In our approach, we train the actor and critic in a MAPPO-like (Yu et al., 2022) manner. Unlike other CTDE model-free approaches that require a global oracle state from the environment, we cannot obtain the oracle state from the world model, and only the predicted observations of each agent are available. To approximate the oracle information in critic training, we enhance each agent's critic with the capability to access the observations of other agents. Since the actor and critic only rely on the reconstructed observations, decoupling from the inner hidden states of the Transformer-based world model, we allow fast inference in the environment without the participation of the world model. It is important for the deployment of policies learned with data-efficient imagination in real-world applications. $\lambda$-target in Dreamer (Hafner et al., 2020) is used to updated the value function. The details of behavior learning objectives and algorithmic description of MARIE are presented in §B and §I, respectively.

## 4 EXPERIMENTS

We consider the most common benchmark – StarCraftII Multi-Agent Challenge (SMAC) (Samvelyan et al., 2019) for evaluating our method. To highlight the sample efficiency brought by model-based imagination, we adopt a low data regime that resembles a similar setting in single-agent Atari domain (Łukasz Kaiser et al., 2020). Additional experiment results on MAMujoco (Peng et al., 2021) (i.e., continuous action space case) is provided in §E.1.

### 4.1 EXPERIMENT SETUP AND EVALUATIONS

**StarCraftII Multi-Agent Challenge.** SMAC (Samvelyan et al., 2019), a suite of cooperative multi-agent environments based on StarCraft II, consists of a set of StarCraft II scenarios. Each scenario depicts a confrontation between two armies of units, one of which is controlled by the built-in game AI and the other by our algorithm. The initial position, number, and type of units in each army

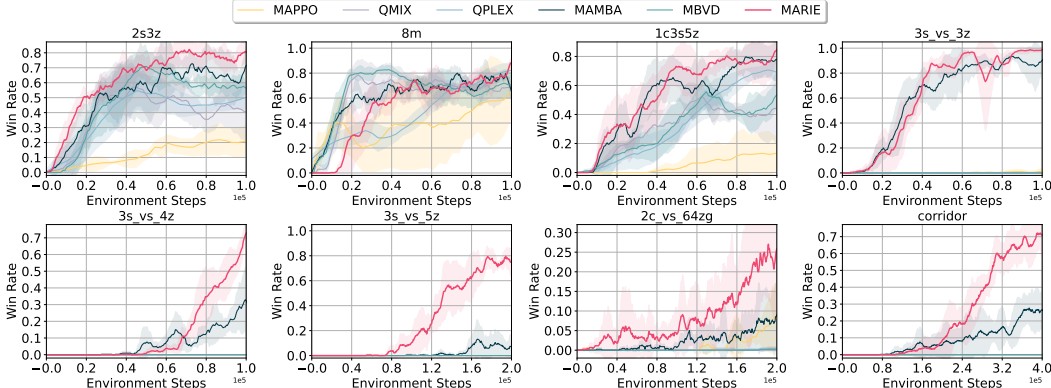

Figure 3: Curves of evaluation win rate for methods in 8 chosen SMAC maps. See Table 1 for win rates. Y axis: win rate; X axis: number of steps taken in the real environment. MARIE demonstrates superior performance and sample efficiency across almost all scenarios.

varies from scenario to scenario, as does the presence or absence of elevated or impassable terrain. And the goal is to win the game within the pre-specified time limit. SMAC emphasizes mastering micromanagement techniques across multiple agents to achieve effective coordination and overcome adversaries. This necessitates both sufficient exploration and appropriate credit assignment for each agent's action. Another notable property of SMAC is that not all actions are accessible during decision-making of each agent, which requires world models to possess an in-depth comprehension of the underlying game mechanics so as to consistently provide valid available action mask estimation within the imagination horizon. Thus, in this benchmark, we additionally add one more head for the prediction of available action mask. During the imagination of MARIE, the available action mask is estimated by this head, instead of being generated manually according to the meaning of each element in the reconstructed observation. The latter introduces too much prior knowledge about StarCraft and can be considered as benchmark hacking.

**Experimental Setup.** We choose 13 representative scenarios from SMAC that includes three levels of difficulty – *Easy*, *Hard*, and *SuperHard*. Specific chosen scenarios can be found in Table 1. In terms of different levels of difficulty, we adopt a similar setting akin to that in (Egorov & Shpilman, 2022) and restrict the number of samples from the real environment to 100k for *Easy* scenarios, 200k for *Hard* scenarios and 400k for *SuperHard* scenarios, to establish a low data regime in SMAC. We compare MARIE with three strong model-free baselines – MAPPO (Yu et al., 2022), QMIX (Rashid et al., 2018) and QPLEX (Wang et al., 2021), and two strong model-based baselines with the same policy learning paradigm as ours – MBVD (Xu et al., 2022) and MAMBA (Egorov & Shpilman, 2022) on SMAC benchmark. Specially, as a multi-agent variant of DreamerV2 (Hafner et al., 2021), MAMBA achieves powerful sample efficiency in various SMAC scenarios via learning in imagination. For each random seed, we compute the win rate across 10 evaluation games at fixed intervals of environmental steps. The hyperparameters of MARIE and other baselines are listed in §D and §H. Particularly, the hyperparameters of model-free baselines in low data regime are directly referred to Egorov & Shpilman (2022) and Liu et al. (2024).

**Main Results.** Overall, we find MARIE achieves significantly better sample efficiency and a higher win rate compared with other strong baselines. We report the averaged win rates over four seeds in Table 1 and provide additional learning curves of several chosen scenarios, shown as Fig. 3. As presented in Table 1 and Fig. 3, MARIE demonstrates superior performance and sample efficiency across almost all scenarios. The improvements in sample efficiency and performance become particularly pronounced with increasing difficulty of scenarios, especially compared to MAMBA that adopts RSSM as the backbone for the world model. We attribute such results to the model capability of the Transformer in local dynamics modeling and global feature aggregation. Benefiting from more powerful strength in modeling sequences, the Transformer-based world model can generate more accurate and consistent imaginations than those relying on the recurrent backbone, which facilitates better policy learning within the imagination of the world model. While the scenarios become harder, e.g. *3s_vs_5z*, our world model can address the challenge of learning more intricate underlying dynamics and further large quantities of accurate imaginations, thereby significantly outperforming other baselines on these scenarios. Moreover, a special scenario *2c_vs_64zg* deserves attention, which

Table 1: Mean evaluation win rate and standard deviation on 13 SMAC maps for different methods over 4 random seeds. We bold the values of the maximum and highlight them with blue color.

| Maps | Difficulty | Steps | MARIE (Ours) | MAMBA (Egorov & Shpilman, 2022) | MAPPO (Yu et al., 2022) | QMIX (Rashid et al., 2018) | QPLEX (Wang et al., 2021) | MBVD (Xu et al., 2022) |
|------|-----------|-------|------|-------|-------|------|-------|------|
| 1c3s5z | | | **85.0**(9.4) | 77.7(15.3) | 18.4(11.0) | 43.6(29.2) | 68.3(7.4) | 60.9(11.4) |
| 2m_vs_1z | | | **95.5**(7.9) | **95.5**(2.3) | 86.7(3.2) | 70.3(14.8) | 84.8(10.8) | 36.7(24.5) |
| 2s_vs_1sc | | | 96.9(7.1) | 95.0(7.1) | **100.0**(0.0) | 0.0(0.0) | 15.7(19.5) | 8.7(14.8) |
| 2s3z | | | **80.5**(9.3) | 71.6(12.7) | 31.2(12.9) | 37.7(15.5) | 50.2(8.4) | 53.4(4.1) |
| 3m | _Easy_ | 100K | **99.5**(0.4) | 87.7(7.1) | 80.5(12.8) | 54.4(22.7) | 88.7(6.9) | 73.9(6.9) |
| 3s_vs_3z | | | **98.9**(1.5) | 89.3(10.1) | 1.2(1.3) | 0.0(0.0) | 0.0(0.0) | 0.0(0.0) |
| 3s_vs_4z | | | **73.0**(6.2) | 29.3(12.3) | 0.0(0.0) | 0.0(0.0) | 0.0(0.0) | 0.0(0.0) |
| 8m | | | **88.0**(3.9) | 65.0(7.7) | 70.3(19.5) | 69.5(12.8) | 83.4(6.4) | 74.7(9.7) |
| MMM | | | **87.6**(3.0) | 50.2(27.6) | 5.5(4.5) | 31.1(17.3) | 69.3(35.1) | 20.5(2.1) |
| so_many_baneling | | | **94.8**(5.9) | 91.6(4.1) | 43.8(15.0) | 20.0(8.9) | 32.2(6.1) | 15.0(10.4) |
| 3s_vs_5z | _Hard_ | 200K | **78.4**(11.2) | 13.4(14.0) | 0.0(0.0) | 0.0(0.0) | 0.0(0.0) | 0.0(0.0) |
| 2c_vs_64zg | | | **25.9**(14.3) | 9.8(8.7) | 7.8(10.2) | 0.5(0.5) | 0.1(0.1) | 0.2(0.4) |
| corridor | _SuperHard_ | 400K | **71.0**(13.8) | 26.5(15.2) | 0.4(0.7) | 0.0(0.0) | 0.0(0.0) | 0.0(0.0) |

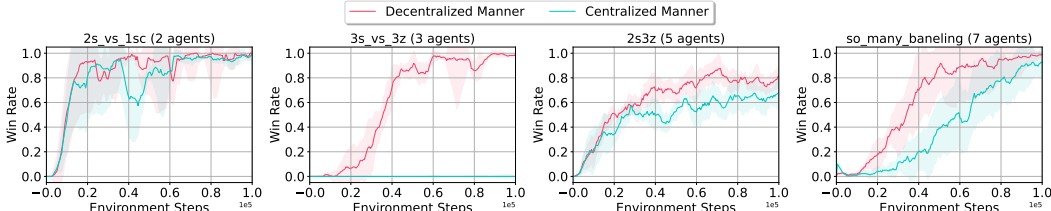

Figure 4: Ablation on what manner to integrate into the design of the world model. _Decentralized_ Manner denotes the standard implementation of MARIE, while _Centralized_ Manner denotes that the world model is designed for learning the joint dynamics of all agents over the joint trajectory. Sample efficiency of the _centralized_ variant encounters a significant drop due to the scalability issue while MARIE is robust to scenarios with various number of agents.

features only 2 agents but with a considerably large action space of up to 70 discrete actions for each agent. Although the performance of MARIE in _2c_vs_64zg_ suffers a relative large variance due to the overly large action space, MARIE achieves a remarkably non-trivial mean win rate just via learning in the imagination. Note that it is easy for the world model to generate ridiculous estimated available action masks without understanding the mechanics behind this scenario, further leading to invalid or even erroneous policy learning in the imaginations of the world model. The performance gap on _2c_vs_64zg_ proves that our Transformer-based world model has higher prediction accuracy and a deeper understanding of the underlying mechanics.

## 4.2 ABLATION STUDIES

**Incorporating CTDE principle with the design of the world model makes MARIE scalable and robust to different number of agents.** We compare our method with a _centralized_ variant of our method, wherein the world model learns the joint dynamics of all agents together over the joint trajectory $\tau = (\ldots, o_t^1, o_t^2, \ldots, o_t^n, a_t^1, a_t^2, \ldots, a_t^n, \ldots)$. Given that $\tau$ already contains the joint observations and actions, we disable the aggregation module in this _centralized_ variant. As illustrated in Figure 4, our comparisons span scenarios involving 2 to 7 agents. When the number of agents is small enough, reducing the multi-agent system to a single-agent one over the joint observation and action space would not cause a prominent scalability issue, as indicated by the result in _2s_vs_1sc_. However, the scalability issue is exacerbated by a growing number of agents. In scenarios featuring more than 3 agents, the sample efficiency of the _centralized_ variant encounters a significant drop, suffering from the exponential surge in spatial complexity of the joint observation-action space. Furthermore, with equal prediction horizons, the parameter amounts in the _centralized_ variant is increased by a factor of 4 or larger. And to achieve the same number of environment steps, the _centralized_ variant demands over twice the original computational time. Instead, with decentralized local dynamics and aggregated global features, MARIE delivers stable and superior sample efficiency.

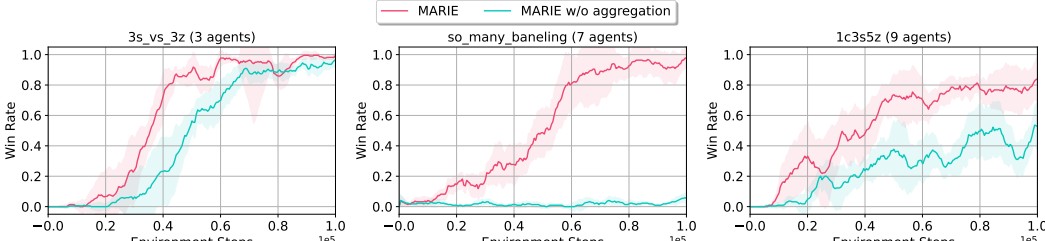

Figure 5: Comparisons between MARIE with and without the usage of the aggregation module. Local dynamics struggles to infer accurate future local observations without agent-wise aggregation.

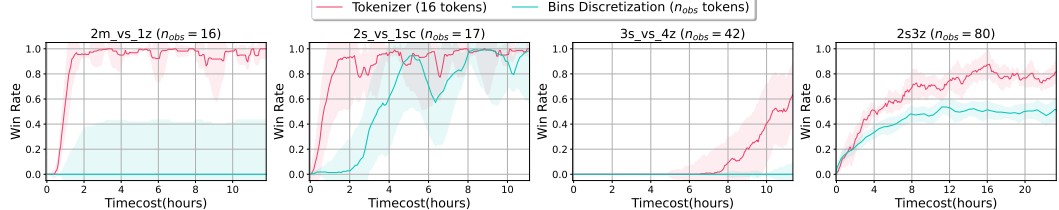

Figure 6: Ablation on the type of discretization for local observations. *Tokenizer* denotes the standard implementation of MARIE; *Bins Discretization* denotes the variant of MARIE where the $n_{obs}$-dimensional observation discretization is performed by projecting the value into one of $m$ fixed-width bins in each dimension independently. X-axis: cumulative run time of algorithms in the same platform. VQ-VAE encapsulates local observations within a succinct sequence of tokens, computationally efficiently promoting the learning of the Transformer-based world model.

**Agent-wise aggregation helps MARIE capture the sophisticated inter-dependency on the generation of each agent's local observation.** To study the influence of agent-wise aggregation, we conduct ablation experiments on the aggregation module over scenarios where the number of agents gradually increases. As shown in Fig. 5, in the 3-agents scenario (e.g., *3s_vs_3z*), the correlation among each agent's local observation tends to be negligible. Therefore, the nearly independent generation of each agent's local observation without any aggregated global feature still leads to performance comparable to that of standard implementation. But as more agents get involved, the inter-dependency becomes dominant. Lacking the global features derived from agent-wise aggregation, the shared Transformer struggles to infer accurate future local observations, thus hindering policy learning in the imaginations of the world model and resulting in notable degradation in the win rate evaluation.

**VQ-VAE encapsulates local observations within a succinct sequence of tokens, promoting the learning of the Transformer-based world model and effectively improving algorithm performance.** Compared to VQ-VAE that discretizes each observation to $K$ tokens from $\mathcal{Z}$, perhaps a more naive tokenizer is projecting the value in each dimension into one of $m$ fixed-width bins (Janner et al., 2021), resulting in a $n_{obs}$-long token sequence for each observation, which we term *Bins Discretization*. We set the number of bins $m$ equal to the size of codebook $|\mathcal{Z}|$ and compare these two types of tokenizers in different environments with various $n_{obs}$. As shown in Fig. 6, the performance of the two tokenizers are comparable only in *2s_vs_1sc* where $n_{obs}$ is close to 16. Even worse, *Bins Discretization* experiences a pronounced decline as $n_{obs}$ increases in more complex environments (e.g., *3s_vs_4z*) under identical training durations. We hypothesize that for a single local observation, a $n_{obs}$-token-long verbose sequence yielded by *Bins Discretization* contains more redundant information compared to VQ-VAE that learns a more compact tokenizer through reconstruction This not only renders the token sequences of *Bins Discretization* obscure and challenging to comprehend, but also results in an increase in model parameter amounts, being more computationally costly. Due to these two factors, *Bins Discretization* exhibits a notably slow convergence. Meanwhile, the result in *2m_vs_1z* indicates *Bins Discretization* may ignore the correlation of different dimensions, which would be helpful in sequence modeling.

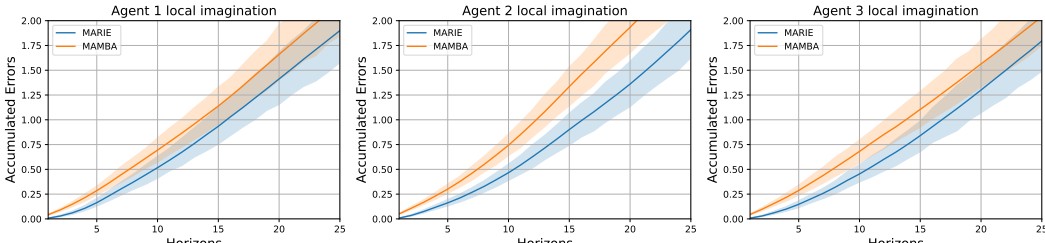

Figure 7: **Compounding model errors.** We compare the imagination accuracy of MARIE to that of MAMBA over the course of a planning horizon in *3s_vs_5z* scenario. MARIE has remarkably better error compounding with respect to prediction horizon than MAMBA.

### 4.3 MODEL ANALYSIS

**Error Accumulation.** A quantitative evaluation of the model's accumulated error versus prediction horizon is provided in Fig. 7. Since learning the world model is tied to a progressively improving policy both in MARIE and MAMBA, we separately use their final policies to sample 10 episodes for fairness. We then compute $L_1$ errors per observation dimension between 1000 trajectory segments randomly sampled from these 20 episodes and their imagined counterpart. The result in Fig. 7 suggests architecture differences play a large role in the world model's long-horizon accuracy. This also provides additional evidence that policy learning can benefit from accurate long-term imaginations, explaining MARIE's notable performance in the *3s_vs_5z* scenario. More precisely, lower generalization error between the estimated dynamics and true dynamics brings a tighter bound between optimal policies derived from these two dynamics according to theoretical results (Janner et al., 2019).

**Attention Patterns.** During model prediction, we delve into the attention maps inside the shared Transformer and the cross attention maps in the Perceiver. Interestingly, we observe two distinct attention patterns involved in the local dynamics prediction. One exhibits a Markovian pattern wherein the observation prediction lays its focus mostly on the previous transition, while the other is regularly striated wherein the model attends to specific tokens in multiple prior transitions. During the agent-wise aggregation, we also identify two distinct patterns – *individuality* and *commonality* among agents. Such diverse patterns in the Transformer and Perceiver may be pivotal for achieving accurate and consistent imaginations of the sophisticated local dynamics. We refer to §C for further details and visualization results.

## 5 CONCLUSION AND LIMITATION

We have introduced a model-based multi-agent algorithm – MARIE, which utilizes a shared Transformer as local dynamic model and a Perceiver as a global agent-wise aggregation module to construct a world model within the multi-agent context. By providing long-term imaginations with policy learning, it significantly boosts the sample efficiency and improves final performance compared to state-of-the-art model-free methods and existing model-based methods with same learning paradigm, in the low data regime. But it should be also noticed that there are potential limitations on the current evaluation on the main experiment with 4 limited seeds, e.g., the limitations of mean and median scores (Agarwal et al., 2021). Thus, we also provide a standardized performance evaluation following the protocol provided by Agarwal et al. (2021) in §G. To further deliver a rigorous statistical validation, evaluation with more seeds is definitely necessary. As the first Transformer-based multi-agent world model for sample-efficient policy learning, we open a new avenue for combining the powerful strength of the Transformer with sample-efficient MARL. Considering the notorious sample inefficiency in multi-agent scenarios, it holds important promise for application in many realistic multi-robot systems, wherein collecting tremendous samples for optimal policy learning is costly and impractical due to the safety. While it has the great potential to bright the future towards achieving smarter multi-agent systems, there still exist limitations in MARIE. For instance, it would suffer from much slower inference speed when used with a very long prediction horizons, due to the auto-regressive property.

## REPRODUCIBILITY STATEMENT

For the implementation details, we provide the detailed instruction in §A. For the practical part, we give experiment setup in §4. The hyper-parameters and implementation details are given in §H. The code will be released publicly after the review process.

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

## A  WORLD MODELS DETAILS AND HYPERPARAMETERS

### A.1  OBSERVATION TOKENIZER

Our tokenizer for local observation discretization is based on the implementation[1] of a vanilla VQ-VAE (van den Oord et al., 2017). Faced with continuous non-vision observation, we build the encoder and decoder as Multi-Layer Perceptrons (MLPs). The decoder is designed with the same hyperparameters as the ones of the encoder. The hyperparameters are listed as Table 2. During the phase of collecting experience from the external environment, each agent takes the reconstructed observations processed by the VQ-VAE as input instead to avoid the distribution shift between policy learning and policy execution.

For training this vanilla VQ-VAE, we use a straight-through estimator to enable gradient backpropagation through the non-differentiable quantization operation in the quantization of VQ-VAE. The loss function for learning the autoencoder is as follows:

$$\mathcal{L}_{\text{VQ}-\text{VAE}}(E, D, \mathcal{Z}) = \mathbb{E}_{i \sim \mathcal{N}} \mathbb{E}_{o^i} \left[ \|o^i - \hat{o}^i\|^2 + \|\text{sg}[E(o^i)] - z_q^i\|^2 + \beta\|\text{sg}[z_q^i] - E(o^i)\|^2 \right] \quad (7)$$

where $\mathcal{N} = \{1, 2, ..., n\}$ denotes the set of agents, $\text{sg}[\cdot]$ denotes the stop-gradient operation and $\beta$ is the coefficient of the commitment loss $\|\text{sg}[z_q^i] - E(o^i)\|^2$. In practice, we found the codebook $\mathcal{Z}$ can suffer from codebook collapse when learning from scratch. Thus, we adopt the Exponential Moving Averages (EMA) (van den Oord et al., 2017) technique to alleviate this problem.

Table 2: VQVAE hyperparameters.

| Hyperparameter | Value |
|---|---|
| **Encoder&Decoder** | |
| Layers | 3 |
| Hidden size | 512 |
| Activation | GELU(Hendrycks & Gimpel, 2016) |
| **Codebook** | |
| Codebook size ($N$) | 512 |
| Tokens per observation ($K$) | 16 |
| Code dimension | 128 |
| Coef. of commitment loss ($\beta$) | 10.0 |

### A.2  TRANSFORMER

The shared Transformer serving as the local dynamics model is based on the implementation of minGPT (Karpathy, 2020). Given a fixed imagination horizon $H$, it first takes a token sequence of length $H(K + 1)$ composed of observation tokens and action tokens, and embeds it into a $H(K + 1) \times D$ tensor via separate embedding tables for observations and actions. Then, the aggregated feature tensor, returned by the agent-wise aggregation module, is inserted after the action embedding tensor at every timestep, forming a final embedding tensor of shape $H(K + 2) \times D$. This tensor is forwarded through fixed Transformer blocks. Here, we adopt GPT2-like blocks (Radford et al., 2019) as the basic blocks. The hyperparameters are listed as Table 3. To enable training across all environments on a single NVIDIA RTX 3090 GPU, we adapt imagination horizon $H$ based on the number of agents.

### A.3  PERCEIVER

The Perceiver (Jaegle et al., 2021) is based on the open-source implementation[2]. By aligning the length of the latent querying array with the number of agents $n$, we obtain the intrinsic global representation feature corresponding to each individual agent. We further dive into the process of agent-wise representation aggregation: (i) the embedding tensor of shape $(K + 1) \times D$ at each timestep, mentioned in Appendix A.2, is concatenated with others from all agents, thereby getting a

---

[1]Code can be found in `https://github.com/lucidrains/vector-quantize-pytorch`

[2]Code can be found in `https://github.com/lucidrains/perceiver-pytorch`

Table 3: Transformer hyperparameters.

| Hyperparameter | Value |
|---|---|
| Imagination horizon ($H$) | $\{15, 8, 5\}$ |
| Embedding dimension | 256 |
| Layers | 10 |
| Attention heads | 4 |
| Weight decay | 0.01 |
| Embedding dropout | 0.1 |
| Attention dropout | 0.1 |
| Residual dropout | 0.1 |

Table 4: Perceiver hyperparameters.

| Hyperparameter | Value |
|---|---|
| Length of latent querying | $n$ (number of agents) |
| Cross attention heads | 8 |
| Inner Transformer layers | 2 |
| Transformer attention heads | 8 |
| Dimension per attention head | 64 |
| Embedding dropout | 0.1 |
| Attention dropout | 0.1 |
| Residual dropout | 0.1 |

$n(K + 1) \times D$ sequence for the joint observation-action pair at the current timestep; (ii) through the cross-attention mechanism with the latent querying array, the original sequence is compressed from length $n(K + 1)$ to $n$; (iii) the compressed sequence is then forwarded through a standard transformer with bidirectional attention inside the Perceiver. The hyperparameters are listed as Table 4.

# B   BEHAVIOUR LEARNING DETAILS

In MARIE, we use MAPPO-like (Yu et al., 2022) actor and critic, where the actor and critic should have been 3-layer MLPs. However, unlike other CTDE model-free approaches, whose critic takes additional global oracle states from the environment in the training phase, our world model hardly provides related predictions in the imagined trajectories. To alleviate this issue, we augment the critic with an attention mechanism and provide it all reconstructed observations $\hat{\boldsymbol{o}}_t$ of all agents. Therefore, the actor $\psi$ remains a 3-layer MLP with ReLU activation, while the critic $\xi$ is enhanced with an extra layer of self-attention, built on top of the original 3-layer MLP, i.e., we overwrite the critic $V_\xi^i(\hat{\boldsymbol{o}}_t) \simeq \mathbb{E}_{\pi_\psi^i}(\sum_{l \geq t} \gamma^{l-t} \hat{r}_l^i)$ for agent $i$. Similar to off-the-shelf CTDE model-free approaches, we adopt parameter sharing across agents.

**Critic loss function**   We utilize $\lambda$-return in Dreamer (Hafner et al., 2020), which employs an exponentially-weighted average of different $k$-steps TD targets to balance bias and variance as the regression target for the critic. Given an imagined trajectory $\{\hat{o}_\tau^i, a_\tau^i, \hat{r}_\tau^i, \hat{\gamma}_\tau^i\}_{t=1}^H$ for agent $i$, $\lambda$-return is calculated recursively as,

$$V_\lambda^i(\hat{\boldsymbol{o}}_t) = \begin{cases} \hat{r}_t^i + \hat{\gamma}_t^i \left[ (1 - \lambda) V_\xi^i(\hat{\boldsymbol{o}}_t) + \lambda V_\lambda^i(\hat{\boldsymbol{o}}_{t+1}) \right] & \text{if} \quad t < H \\ V_\xi^i(\hat{\boldsymbol{o}}_t) & \text{if} \quad t = H \end{cases} \tag{8}$$

The objective of the critic $\xi$ is to minimize the mean squared difference $\mathcal{L}_\xi^i$ with $\lambda$-returns over imagined trajectories for each agent $i$, as

$$\mathcal{L}_\xi^i = \mathbb{E}_{\pi_\psi^i} \left[ \sum_{t=1}^{H-1} \left( V_\xi^i(\hat{\boldsymbol{o}}_t) - \text{sg}(V_\lambda^i(\hat{\boldsymbol{o}}_t)) \right)^2 \right] \tag{9}$$

where $\text{sg}(\cdot)$ denotes the stop-gradient operation. We optimize the critic loss with respect to the critic parameters $\xi$ using the Adam optimizer.

Table 5: Behaviour learning hyperparameters.

| Hyperparameter | Value |
|---|---|
| Imagination Horizon ($H$) | $\{15, 8, 5\}$ |
| Predicted discount label $\gamma$ | 0.99 |
| $\lambda$ | 0.95 |
| $\eta$ | 0.001 |
| Clipping parameter $\epsilon$ | 0.2 |

**Actor loss function**    The objective for the action model $\pi_\psi(\cdot|\hat{o}_t^i)$ is to output actions that maximize the prediction of long-term future rewards made by the critic. To incorporate intermediate rewards more directly, we train the actor to maximize the same $\lambda$-return that was computed for training the critic. In terms of the non-stationarity issue in multi-agent scenarios, we adopt PPO updates, which introduce important sampling for actor learning. The actor loss function for agent $i$ is:

$$\mathcal{L}_\psi^i = -\mathbb{E}_{p_\phi, \pi_{\psi_{old}}^i} \Big[ \sum_{t=0}^{H-1} \min \Big( r_t^i(\psi) A_t^i, \mathrm{clip}(r_t^i(\psi), 1-\epsilon, 1+\epsilon) A_t^i \Big) + \eta \mathcal{H}(\pi_\psi^i(\cdot|\hat{o}_t^i)) \Big] \quad (10)$$

where $r_t^i(\psi) = \pi_\psi^i/\pi_{\psi_{old}}^i$ is the policy ratio and $A_t^i = \mathrm{sg}(V_\lambda^i(\hat{\boldsymbol{o}}_t) - V_\xi^i(\hat{\boldsymbol{o}}_t))$ is the advantage. We optimize the actor loss with respect to the actor parameters $\psi$ using the Adam optimizer. In the discount prediction of MARIE, we set its learning target $\gamma$ to be 0.99. Overall hyperparameters are shown in Table 5.

## C    EXTENDED ANALYSIS ON ATTENTION PATTERNS

To provide qualitative analysis of our world model, we select typical scenarios – *3s_vs_5z* where our method achieves the most significant improvement compared to other baselines for visualizing attention maps inside the Transformer. For the sake of simple and clear visualization, we set the imagination horizon $H$ as 5. In terms of cross-attention maps in the aggregation module, we select a scenario *2s3z* including 5 agents for visualization. Visualization results are depicted as Fig. 8 and Fig. 9.

The prediction of local dynamics entails two distinct attention patterns. The left one in Fig. 8 can be interpreted as a Markovian pattern, in which the observation prediction lays its focus on the previous transition. In contrast, the right one is regularly striated, with the model attending to specific tokens in multiple prior observations. In terms of the agent-wise aggregation, we also identify two distinct patterns: *individuality* and *commonality*. The top one in Fig. 9 illustrates that each agent flexibly attends to different tokens according to their specific needs. In contrast, the bottom one exhibits consistent attention allocation across all agents, with attention highlighted in nearly identical positions. The diverse patterns in the Transformer and Perceiver may be the key to accurate and consistent imagination.

## D    BASELINE IMPLEMENTATION DETAILS

**MAMBA** (Egorov & Shpilman, 2022) is evaluated based on the open-source implementation: `https://github.com/jbr-ai-labs/mamba` with the hyperparameters in Table 6.

**MAPPO** (Yu et al., 2022) is evaluated based on the open-source implementation: `https://github.com/marlbenchmark/on-policy` with the common hyperparameters in Table 7.

**QMIX** (Rashid et al., 2018) is evaluated based on the open-source implementation: `https://github.com/oxwhirl/pymarl` with the hyperparameters in Table 8.

**QPLEX** (Wang et al., 2021) is evaluated based on the open-source implementation: `https://github.com/wjh720/QPLEX` with the hyperparameters in Table 9. Since its implementation is mostly based on the open-source implementation: PyMARL (Samvelyan et al., 2019), its most hyperparameters setting remains the same as the one in QMIX in addition to its own special hyperparameters.

Table 6: Hyperparameters for MAMBA in SMAC environments.

| Hyperparameter | Value |
|---|---|
| Batch size | 256 |
| $\lambda$ for $\lambda$-return computation | 0.95 |
| Entropy coefficient | 0.001 |
| Entropy annealing | 0.99998 |
| Number of policy updates | 4 |
| Epochs per policy update | 5 |
| Clipping parameter $\epsilon$ | 0.2 |
| Actor Learning rate | 0.0005 |
| Critic Learning rate | 0.0005 |
| Discount factor $\gamma$ | 0.99 |
| Model Learning rate | 0.0002 |
| Number of model training epochs | 60 |
| Number of imagined rollouts | 800 |
| Sequence length | 20 |
| Imagination horizon $H$ | 15 |
| Buffer size | $2.5 \times 10^5$ |
| Number of categoricals | 32 |
| Number of classes | 32 |
| KL balancing entropy weight | 0.2 |
| KL balancing cross entropy weight | 0.8 |
| Gradient clipping | 100 |
| Collected trajectories between updates | 1 |
| Hidden size | 256 |

Table 7: Common hyperparameters for MAPPO in SMAC environments.

| Hyperparameter | Value |
|---|---|
| Batch size | num envs × buffer length × num agents |
| Mini batch size | batch size / mini-batch |
| Recurrent data chunk length | 10 |
| GAE $\lambda$ | 0.95 |
| Discount factor $\gamma$ | 0.99 |
| Value loss | huber loss |
| Huber delta | 10.0 |
| Optimizer | Adam |
| Optimizer learning rate | 0.0005 |
| Optimizer epsilon | $1 \times 10^{-5}$ |
| Weight decay | 0.0 |
| Gradient clipping | 10 |
| Network initialization | orthogonal |
| Use reward normalization | True |
| Use feature normalization | True |

Table 8: Hyperparameters for QMIX in SMAC environments.

| Hyperparameter | Value |
|---|---|
| Batch size | 32 |
| Buffer size | 5000 |
| Epsilon in epsilon-greedy | $1.0 \rightarrow 0.05$ |
| Epsilon anneal time | 50000 |
| Train interval | 1 episode |
| Discount factor $\gamma$ | 0.99 |
| Optimizer | RMSProp |
| RMSProp $\alpha$ | 0.99 |
| RMSProp $\epsilon$ | $10^{-5}$ |
| Gradient clipping | 10 |

Table 9: Hyperparameters for QPLEX in SMAC environments.

| Hyperparameter | Value |
|---|---|
| Batch size | 32 |
| Buffer size | 5000 |
| Epsilon in epsilon-greedy | $1.0 \rightarrow 0.05$ |
| Epsilon anneal time | 50000 |
| Train interval | 1 episode |
| Discount factor $\gamma$ | 0.99 |
| Optimizer | RMSProp |
| RMSProp $\alpha$ | 0.99 |
| RMSProp $\epsilon$ | $10^{-5}$ |
| Gradient clipping | 10 |
| Number of layers in HyperNetwork | 1 |
| Number of heads in the attention module | 4 |

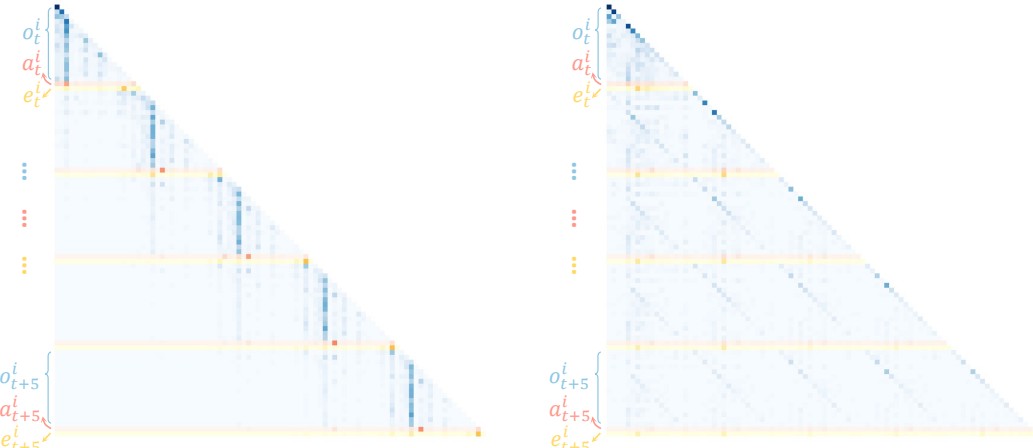

Figure 8: **Attention patterns in the Transformer.** We observe two distinct types of attention weights during the prediction of local dynamics. In the first one (***left***), the next observation prediction is primarily dependent on the last transition, which means the world model has learned the Markov property corresponding to Dec-POMDPs. The second type (***right***) exhibits a regularly striated pattern, where the next observation prediction hinges overwhelmingly on the same dimension of multiple previous timesteps. The above attention weights are produced by a sixth-layer and ninth-layer attention head during imaginations on the *3s_vs_5z* scenario.

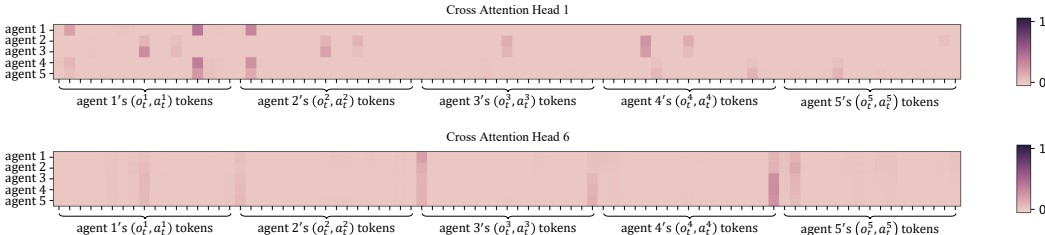

Figure 9: **Cross attention patterns in the Perceiver.** We observe the *individuality* and *commonality* in the agent-wise aggregation. The top part of the figure represents the *individuality*, where agents adjust their attentions over the whole joint token sequence at timestep $t$ flexibly according to their own needs. In contrast, the bottom exhibits the *commonality*, where every agent's attention over the joint token sequence is emphasized in the similar positions of the sequence. The cross attention weights mentioned above are produced by the first and sixth head of the cross attention within the Perceiver, during the agent-wise aggregation on the *2s3z* scenario.

**MBVD** (Xu et al., 2022) is evaluated based on the implementation in its supplementary material from https://openreview.net/forum?id=flBYpZkW6ST with the hyperparameters in Table 10. Akin to QPLEX, its implementation is based on the open-source implementation: PyMARL, its most hyperparameters setting remains the same as the one in QMIX in addition to its own special hyperparameters.

# E ADDITIONAL EXPERIMENTS

## E.1 EVALUATIONS ON MAMUJOCO

The Multi-Agent MuJoCo (MAMuJoCo) (Peng et al., 2021) environment is a multi-agent extension of MuJoCo. While the MuJoCo tasks challenge a robot to learn an optimal way of motion, MAMuJoCo models each part of a robot as an independent agent — for example, a leg for a spider or an arm for a swimmer — and requires the agents to collectively perform efficient motion. With the increasing variety of the body parts, MAMujoco can be also considered as a testbed for evaluating the coordination among heterogeneous agents, which poses a big challenge for learning the multi-agent dynamics inside it, especially in a *decentralized* manner.

Table 10: Hyperparameters for MBVD in SMAC environments.

| Hyperparameter | Value |
|---|---|
| Batch size | 32 |
| Buffer size | 5000 |
| Epsilon in epsilon-greedy | $1.0 \rightarrow 0.05$ |
| Epsilon anneal time | 50000 |
| Train interval | 1 episode |
| Discount factor $\gamma$ | 0.99 |
| Optimizer | RMSProp |
| RMSProp $\alpha$ | 0.99 |
| RMSProp $\epsilon$ | $10^{-5}$ |
| Gradient clipping | 10 |
| Number of layers in HyperNetwork | 1 |
| Number of heads in the attention module | 4 |
| Horizon of the imagined rollout | 3 |
| KL balancing $\alpha$ | 0.3 |
| Dimension of the latent state $\hat{s}$ | num agents x 16 |

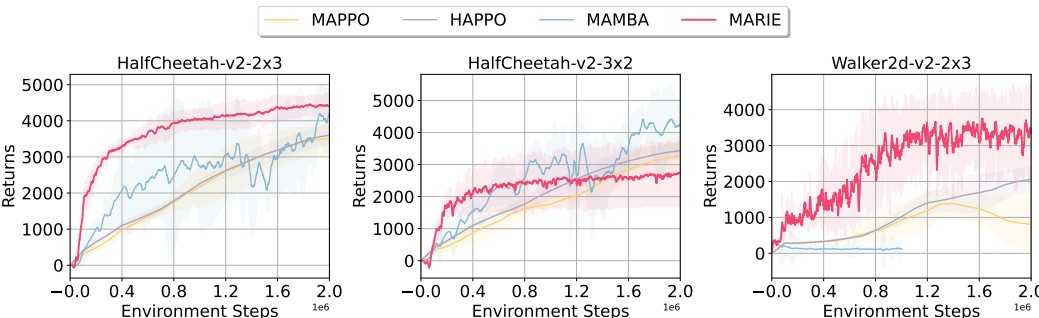

Figure 10: Curves of the performance for MARIE, MAMBA, HAPPO and MAPPO in 3 chosen MAMujoco scenarios. Y axis: return; X axis: number of steps taken in the real environment.

While MAMBA (Egorov & Shpilman, 2022) originally does not take the continuous action space case into consideration, which is a obvious limitation in it, we would like to evaluate MARIE in such case, e.g., MAMujoco, to better demonstrate that our method can be also effective in other multi-agent domains. In MAMujoco, we discretize the scalar in each dimension of continuous actions into one of 256 fixed-width bins independently to obtain discrete action tokens for local dynamics learning. As the behaviour learning in MARIE adopts a MAPPO-like and on-policy manner, we choose two strong on-policy PPO-based baselines – MAPPO (Yu et al., 2022) and HAPPO (Kuba et al., 2021). Additionally, we also include MAMBA as a model-based baseline for comparison. Since MAMBA(Egorov & Shpilman, 2022) was originally originally designed for domains with discrete action space, significant effort was required to adapt and evaluate it on MAMujoco, which features continuous action space. The experiments are conducted in *HalfCheetah-v2-2x3*, *HalfCheetah-v2-3x2* and *Walker2d-v2-2x3*. The learning curves of the return averaged over 4 seeds are presented as Figure 10. Notably, MAMBA fails to enhance policy learning in the *Walker2d-v2-2x3* scenario and remains exceptionally time-consuming. Consequently, we report its results only for 1 million environment steps in this scenario.

As illustrated in Figure 10, our MARIE consistently shows superior sample efficiency and achieves the best performance in 2 of 3 scenarios with limited 2M environment steps. For the performance difference between MAMBA and MARIE in *HalfCheetah-v2-3x2*, we hypothesize that MAMBA's policy learning benefits significantly from using the internal recurrent features of the world model as inputs in this scenario, while the policy in our method only takes the reconstructed observation as input in order to support fast deployment in the environment without the participation of the world model. We attribute the performance gap between MARIE and other two model-free baselines in *HalfCheetah-v2-3x2* to the access to global oracle state in the chosen baselines. The policy in our

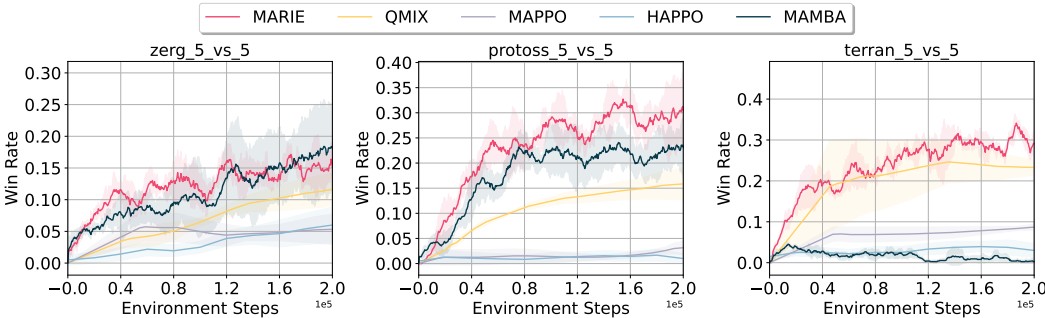

Figure 11: Curves of the performance for MARIE, MAMBA, QMIX, HAPPO and MAPPO in 3 chosen SMACv2 scenarios. Y axis: Win Rate; X axis: number of steps taken in the real environment. While MARIE shows competitive performance to MAMBA on zerg_5_vs_5, MARIE is superior to all other baselines in terms of sample efficiency and final performance in the rest 2 scenarios.

algorithm is purely learned from the inner imaginations of the world model where there is only reconstructed local observation. Considering MAMujoco is a multi-heterogeneous-agent benchmark which necessitates a more precise credit assignment during training, it would be much more helpful for policy learning to have access to the true global oracle state than in other benchmarks. But overall, our MARIE presents a faster convergence rate, implying that our Transformer-based world model can generate accurate imaginations and bring remarkable sample efficiency.

### E.2 EVALUATIONS ON SMACv2

Given known serious flaws in SMACv1 (e.g., the tricky open-loop policy issue), we extend our evaluation of MARIE to SMACv2 (Ellis et al., 2023), which introduces more stochasticity and partial observability. In this comparison, we benchmark MARIE against four baselines: MAPPO, HAPPO, QMIX and MAMBA. For each random seed, we adopt the same evaluation protocol as the main experiment on SMACv1. Importantly, the hyperparameters of MARIE remain unchanged, as detailed in §H. Here, we directly use the results of MAPPO and QMIX provided in the official SMACv2 repository[3]. Illustrated in Figure 11, while MARIE shows competitive performance to MAMBA on zerg_5_vs_5, MARIE is superior to all other baselines in terms of sample efficiency and final performance in the rest 2 scenarios.

### E.3 COMPARISON WITH QMIX AND QPLEX WITH DIFFERENT EPSILON ANNEALING TIME

Considering the potentially inappropriate influence of a large $\epsilon$ annealing time used in the epsilon-greedy algorithm when evaluated in the low data regime evaluation, we run QMIX and QPLEX with a smaller $\epsilon$ annealing time, and compare the performance of them with ours. The result is reported in Table 11. The reported result shows that the original hyperparameters used in the main experiment, which are also directly referred to Egorov & Shpilman (2022) and Liu et al. (2024), are reasonable since the performance of QMIX and QPLEX under the original hyperparameters is superior to the ones with a smaller $\epsilon$ annealing time at most scenarios. Besides, our MARIE still consistently outperforms QMIX and QPLEX with a smaller $\epsilon$ annealing time.

### E.4 COMPARISON WITH EXISTING TRANSFORMER-BASED WORLD MODELS

Existing Transformer-based world models are primarily designed for single-agent scenarios, but they can be naturally adapted to multi-agent settings, modeling either independently local dynamics or joint dynamics. Fortunately, we have included IRIS as a Transformer-based world model baseline in our ablation experiments. Specifically, the *Centralized Manner* and *MARIE w/o aggregation* variants from our ablation experiments correspond to IRIS baseline variants under different deployment strategies. But different from their original implementation, these IRIS baseline variants also uses the same actor-critic method as MARIE during learning in imaginations phase (i.e., using PPO instead

---

[3]Results of QMIX and MAPPO are available at `https://github.com/oxwhirl/smacv2/tree/main/smacv2/examples/results`.

Table 11: Mean evaluation win rate and standard deviation for QMIX and QPLEX with different epsilon anneal time $t_\epsilon$ over 4 random seeds. We bold the values of the maximum.

| Maps | Steps | MARIE | QMIX ($t_\epsilon = 50000$) | QMIX ($t_\epsilon = 10000$) | QPLEX ($t_\epsilon = 50000$) | QPLEX ($t_\epsilon = 10000$) |
|---|---|---|---|---|---|---|
| 1c3s5z | | **85.0**(9.4) | 43.6(29.2) | 33.3(15.0) | 68.3(7.4) | 44.8(11.0) |
| 2m_vs_1z | | **95.5**(7.9) | 70.3(14.8) | 36.1(28.2) | 84.8(10.8) | 93.2(4.7) |
| 2s_vs_1sc | | **96.9**(7.1) | 0.0(0.0) | 3.9(6.7) | 15.7(19.5) | 43.2(32.4) |
| 2s3z | 100K | **80.5**(9.3) | 37.7(15.5) | 29.1(20.3) | 50.2(8.4) | 28.3(11.5) |
| 3m | | **99.5**(0.4) | 54.4(22.7) | 63.8(14.6) | 88.7(6.9) | 85.0(11.3) |
| 3s_vs_3z | | **98.9**(1.5) | 0.0(0.0) | 0.0(0.0) | 0.0(0.0) | 0.0(0.0) |
| 3s_vs_4z | | **73.0**(6.2) | 0.0(0.0) | 0.0(0.0) | 0.0(0.0) | 0.0(0.0) |
| 8m | | **88.0**(3.9) | 69.5(12.8) | 68.6(13.6) | 83.4(6.4) | 79.7(9.8) |
| MMM | | **87.6**(3.0) | 31.1(17.3) | 18.9(4.3) | 69.3(35.1) | 20.2(7.7) |
| so_many_baneling | | **94.8**(5.9) | 20.0(8.9) | 30.7(18.5) | 32.2(6.1) | 37.7(9.2) |
| 2c_vs_64zg | 200K | **25.9**(14.3) | 0.5(0.5) | 0.0(0.0) | 0.1(0.1) | 0.0(0.0) |
| 3s_vs_5z | | **78.4**(11.2) | 0.0(0.0) | 0.0(0.0) | 0.0(0.0) | 0.0(0.0) |
| corridor | 400K | **71.0**(13.8) | 0.0(0.0) | 0.0(0.0) | 0.0(0.0) | 0.0(0.0) |

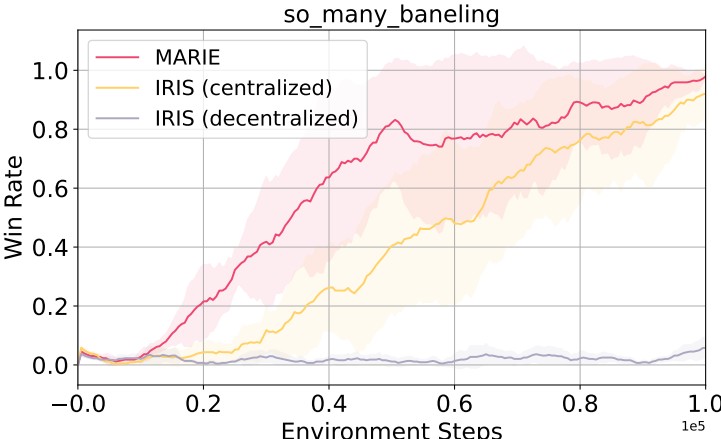

Figure 12: Comparison with two direct extension of IRIS in the *so_many_baneling* scenario.

of REINFORCE for behaviour learning). With a shared behaviour learning phase, we can analyze the reason why existing single-agent Transformer-based world model cannot be directly adapted to MARL. As shown in Figure 12, without incorporating CTDE principle, the learning of single-agent world model would be disrupted by the scalability and non-stationarity issues.

### E.5 COMPARISON AGAINST MAMBA WITH DIFFERENT IMAGINATION HORIZON

We report the performance of MARIE and MAMBA on *3s_vs_5z* with different imagination horizon in Table 12. And the result shows that the performance gap between the two is not related to the choice of imagination horizon. Interestingly, a larger imagination horizon may help policy learning in imagination. We hypothesize that longer imagined trajectories help alleviating shortsighted behaviours in policy learning.

Table 12: The performance of MARIE and MAMBA on *3s_vs_5z* with different imagination horizon.

| Maps | Method | Horizon $H = 8$ | Horizon $H = 15$ | Horizon $H = 25$ |
|---|---|---|---|---|
| 3s_vs_5z | MARIE | $0.40 \pm 0.34$ | $0.75 \pm 0.09$ | $0.78 \pm 0.11$ |
| | MAMBA | $0.00 \pm 0.00$ | $0.13 \pm 0.14$ | $0.16 \pm 0.13$ |

## F   ADDITIONAL DISCUSSION BETWEEN CODREAMER AND MARIE

Additionally, a recent method CoDreamer (Toledo & Prorok, 2024) extends DreamerV3 (Hafner et al., 2023) to the multi-agent setting, using GAT V2 (Brody et al., 2021) for communication among agents' world models and policies. Though the aggregation modules in CoDreamer and ours are both built upon the Transformer architecture, our focus lies in computational efficiency of aggregation while it focuses on the underlying topological graph structure among agents. However, a fundamental difference is the backbone used for modeling the local dynamics. While we cast the local dynamics learning as the sequence modeling over discrete tokens, which can be achieved by using auto-regressive Transformers with causal attention mechanism, CoDreamer directly adopts the RSSM framework in DreamerV3.

## G   STANDARDIZED PERFORMANCE EVALUATION PROTOCOL

Agarwal et al. (2021) discuss the limitations of mean and median scores, and show that substantial discrepancies arise between standard point estimates and interval estimates in RL benchmarks. To deliver a rigorous statistical evaluation, we summarize in Figure 13 the win rate with stratified bootstrap confidence intervals for mean, median, and inter-quartile mean (IQM). For finer comparisons, we also provide probabilities of improvement in Figure 14.

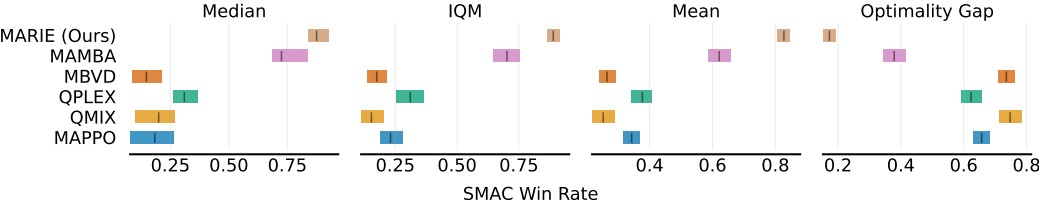

Figure 13: Mean, median, and inter-quartile mean win rate, computed with stratified bootstrap confidence intervals. 4 runs for all algorithms.

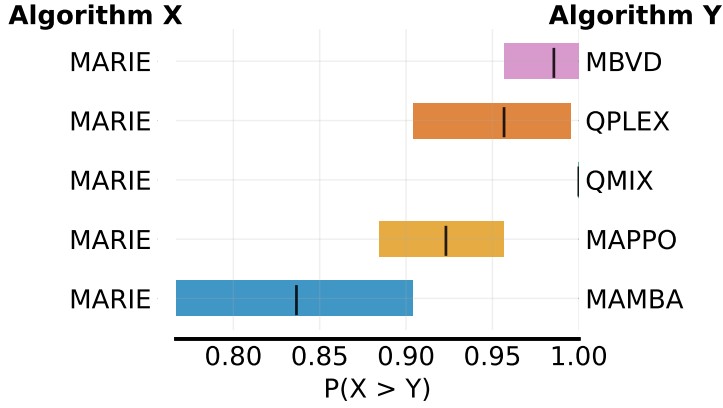

Figure 14: Probabilities of improvement (Agarwal et al., 2021).

Table 13: Hyperparameters for MARIE in SMAC environments.

| Hyperparameter | Value |
|---|---|
| Batch size for tokenizer training | 256 |
| Batch size for world model training | 30 |
| Optimizer for tokenizer | AdamW |
| Optimizer for world model | AdamW |
| Optimizer for actor & critic | Adam |
| Tokenizer learning rate | 0.0003 |
| World model learning rate | 0.0001 |
| Actor learning rate | 0.0005 |
| Critic learning rate | 0.0005 |
| Gradient clipping for actor & critic | 100 |
| Gradient clipping for tokenizer | 10 |
| Gradient clipping for world model | 10 |
| Weight decay for world model | 0.01 |
| $\lambda$ for $\lambda$-return computation | 0.95 |
| Discount factor $\gamma$ | 0.99 |
| Entropy coefficient | 0.001 |
| Buffer size (transitions) | $2.5 \times 10^5$ |
| Number of tokenizer training epochs | 200 |
| Number of world model training epochs | 200 |
| Collected transitions between updates | $\{100, 200\}$ |
| Epochs per policy update (PPO epochs) | 5 |
| PPO Clipping parameter $\epsilon$ | 0.2 |
| Number of imagined rollouts | 600 or 400 |
| Imagination horizon $H$ | $\{15, 8, 5\}$ |
| Number of policy updates | $\{4, 10, 30\}$ |
| Number of stacking observations | 5 |
| Observe agent id | False |
| Observe last action of itself | False |

Table 14: Computational time consumption of MARIE in SMAC.

| Environment Steps | 100000 | 200000 | 400000 |
|---|---|---|---|
| Training Time | 1 day | 2-3 days | 4 days |
| Usage of GPU Mem | 22GB | 22GB | 22GB |

## H  PARAMETERS SETTING AND COMPUTATIONAL CONSUMPTION OF MARIE

All our experiments are run on a machine with a single NVIDIA RTX 3090 GPU, a 36-core CPU, and 128GB RAM. We provide the hyperparameters of MARIE for experiments in SMAC, shown as Table 13. To enable the running of experiments in all SMAC scenarios with a single NVIDIA RTX 3090 GPU, we set the imagination horizon $H$ as 8 for other scenarios involving the number of agents $n > 5$, 15 for $n \leq 5$. In *so_many_baneling* and *2s3z*, we set the imagination horizon $H$ as 5. Correspondingly, the number of policy updates in imaginations varies with imagination horizon $H$. As for the scenario *2c_vs_64zg*, considering the significantly large action space in it, we enable the observation of agent id and last action for each agent and disable stacking the last 5 observations as input to the policy.

Based on the above reported setting, we present a rough computational consumption in Table 14.

# I Overview of MARIE Algorithm

Pseudo-code is summarized as Algorithm 1.

---
**Algorithm 1** MARIE

---
// main loop of training
**for** *epochs* **do**
    collect_experience(*num_transitions*)
    **for** *learning_world_model_steps_per_epoch* **do**
        train_world_model()
    **end for**
    **for** *learning_behaviour_steps_per_epoch* **do**
        train_agents()
    **end for**
**end for**

**function** collect_experience($n$):
$\boldsymbol{o}_0 \leftarrow$ env.reset()
**for** $t = 0, \ldots, n-1$ **do**
    // processed by VQ-VAE
    $\hat{\boldsymbol{o}}_t \leftarrow D(E(\boldsymbol{o}_t))$
    Sample $a_t^i \sim \pi_\psi^i(a_t^i | \hat{o}_t^i), \forall i$
    $\boldsymbol{o}_{t+1}, r_t, done \leftarrow$ env.step($\boldsymbol{a}_t$)
    **if** $done = True$ **then**
        $\boldsymbol{o}_{t+1} \leftarrow$ env.reset()
        $\gamma_t \leftarrow 0.$
    **else**
        $\gamma_t \leftarrow 0.99$
    **end if**
**end for**
$\mathcal{D} \leftarrow \mathcal{D} \cup \{\boldsymbol{o}_t, \boldsymbol{a}_t, r_t, \gamma_t\}_{t=0}^{n-1}$

**function** train_world_model():
Sample $\{\boldsymbol{o}_t, \boldsymbol{a}_t, r_t, \gamma_t\}_{t=\tau}^{t=\tau+H-1}$
Update $(E, D, \mathcal{Z})$ via $\mathcal{L}_{\text{VQ-VAE}}$ over observations $\{\boldsymbol{o}_t\}_{t=\tau}^{t=\tau+H-1}$
**for** agent $i = 1, \ldots, n$ **do**
    Update $\phi, \theta$ via $\mathcal{L}_{\text{Dyn}}(\phi, \theta)$ over local trajectories $\{o_t^i, a_t^i, r_t, \gamma_t\}_{t=\tau}^{t=\tau+H-1}$
**end for**

**function** train_agents():
Sample an initial observation $\boldsymbol{o}_0 \sim \mathcal{D}$
$\{x_{0,j}^i\}_{j=1}^K \leftarrow E(o_0^i), \hat{o}_0^i \leftarrow D(E(o_0^i)), \forall i$
**for** $t = 0, \ldots, H-1$ **do**
    Sample $a_t^i \sim \pi_\psi^i(a_t^i | \hat{o}_t^i), \forall i$
    Aggregate $(x_{t,1}^1, \ldots, x_{t,K}^1, a_t^1, \ldots, x_{t,1}^n, \ldots, x_{t,K}^n, a_t^n)$ into $(e_t^1, \ldots, e_t^n)$ via the Perceiver $\theta$
    Sample $\hat{x}_{t+1,\cdot}^i, \hat{r}_t^i, \hat{\gamma}_t^i \sim p_\phi(\hat{x}_{t+1,\cdot}^i, \hat{r}_t^i, \hat{\gamma}_t^i | x_{0,\cdot}^i, a_0^i, e_0^i, \ldots, \hat{x}_{t,\cdot}^i, a_t^i, \hat{e}_t^i), \forall i$
    $\hat{o}_{t+1}^i \leftarrow D(\hat{x}_{t+1,\cdot}^i), \forall i$
**end for**
**for** agent $i = 1, \ldots, n$ **do**
    Update actor $\pi_\psi^i$ and critic $V_\xi^i$ via $\mathcal{L}_{\text{Dyn}}(\phi, \theta)$ over imagined trajectories $\{\hat{o}_t^i, a_t^i, \hat{r}_t^i, \hat{\gamma}_t^i\}_{t=0}^{t=H-1}$
**end for**

---

