# OpenReview forum: "Decentralized Transformers with Centralized Aggregation are Sample-Efficient Multi-Agent World Models"
_ICLR.cc/2025/Conference — Submitted to ICLR 2025_

### Official Review · Reviewer_Cbz9 · 2024-10-21

**Soundness:** 2
**Presentation:** 3
**Contribution:** 2
**Rating:** 5
**Confidence:** 5

**Summary:**

The paper introduces MARIE (Multi-Agent auto-Regressive Imagination for Efficient learning), a Transformer-based architecture designed to improve sample efficiency through improving the accuracy of multi-agent world modelling. The authors aim to address challenges of world modelling in MARL, particularly the scalability and non-stationarity issues, by using decentralised local dynamics combined with centralised aggregation through a Perceiver Transformer. The architecture/algorithm is evaluated on SMAC and additional experiments are conducted on MAMujoco, showing improved sample efficiency and overall performance compared to existing model-free and model-based methods.

**Strengths:**

- The idea is presented clearly, and the architecture and experimental setup are detailed well. The integration of decentralised and centralised components is relatively straightforward and understandable.

- The submission provides comprehensive implementation details and references all the open-source repositories used for baselines, making it likely reproducible. The authors also mention that code will be released after the review process, which supports transparency.

- The paper presents results that show improvements over baselines. The usage of SMAC and MAMujoco environments offers a broad view of the architecture’s capabilities i.e. both discrete and continuous action spaces.

- The introduction of a Perceiver Transformer for centralised aggregation in a multi-agent context is an interesting approach and could provide valuable insights for the community.

**Weaknesses:**

- The experiments lack rigourous statistical testing, which is critical given the limited number of seeds (only four). This raises concerns about the reliability and significance of the results. Referring to rigorous RL evaluation protocols such as those outlined in [rliable_code](https://github.com/google-research/rliable), [rliable_paper](https://arxiv.org/pdf/2108.13264) and [marl_eval](https://proceedings.neurips.cc/paper_files/paper/2022/file/249f73e01f0a2bb6c8d971b565f159a7-Paper-Conference.pdf) (among others) would have strengthened the empirical claims. These evaluation protocols have become a common standard that the community should uphold. Without statistical validation, it's hard to confirm that the reported improvements are statistically significant.

- The use of SMAC, particularly SMAC v1, is problematic as it is an outdated benchmark with known serious flaws, see [SMACv2](https://arxiv.org/abs/2212.07489). The evaluation would benefit from using the updated SMAC v2 version, which addresses some of these issues and gives more credibility to the method. Furthermore, the absence of comparisons with MAMBA in environments beyond SMAC makes it difficult to comprehensively evaluate the advantages of MARIE over existing architectures.

- While the architecture is interesting, the novelty might be overstated. There are similarities between MARIE and existing methods, such as MAMBA, with no stark performance difference (at least without the statistical testing, i don't believe we can make a fair claim that the difference is stark). Additionally, a recent approach, see https://openreview.net/forum?id=f2bgGy7Af7,  using graph attention networks (GATv2) (which are essentially transformers in a sense) closely mirrors the methodology, questioning the novelty of MARIE's transformer-based aggregation. Not mentioning this in the related work section detracts from the contribution's novelty marginally. However, i will say this is the least important weak point considering that if the results were rigorously validated i would still believe this methodology is worth it for the community to see.

**Questions:**

### Questions

1. Why were only four seeds used for evaluation, and are there plans to conduct more extensive statistical testing to validate the results? I understand that experiments can be expensive to run and lots of tasks were used thus I thoroughly recommend reading the RLiable paper to see how you can address this without increasing computational budget. I believe you could simply run the tests with existing results.

2. Have you considered using SMAC v2, or is there a rationale for continuing with SMAC v1 despite its known flaws?

3. Could you provide a direct comparison on environments beyond SMAC for methods like MAMBA and potentially CoDreamer?

4. Given that the difference in compounding error of the world models between MAMBA and MARIE get worse over time, would the performance gap between the two be reduced if using a smaller imagination horizon for training and does this possibly bridge the gap?

### Suggestions:

- **Use of Benchmarks**: Including evaluations on more diverse environments, such as those in the updated SMAC version or other cooperative multi-agent benchmarks, would strengthen the paper’s claims.

- **Statistical Validation**: Incorporating more seeds and employing robust statistical methods would add credibility to the results.

In conclusion, I don’t feel confident enough that the results presented truly indicate a statistically significant performance improvement and that the architecture itself doesn’t provide enough of a difference to warrant acceptance without the solid empirical proof that it is a superior method. If a) the RLiable evaluation methodology is run and the results present statistically significant improvements and b) MAMBA is run on MAMujoco with similar conclusions to SMAC then i will be willing to raise my score.

---

> ### Author Response · Authors · 2024-11-22
> **Response to Reviewer Cbz9**
>
> Dear Reviewer Cbz9,
>
> We sincerely appreciate your precious time and constructive comments. Below, we provide a detailed point-by-point response, and we hope our clarifications and revision adequately address your concerns.
>
> **W1 & Q1**: Lack of standardized performance evaluation protocol given the limited number of seeds.
>
> **Response**: Thanks for the valuable comment! Upon the reviewer's suggestion, we summarize in Figure 13 the win rate with stratified bootstrap confidence intervals for mean, median, and interquartile mean (IQM). For finer comparisons, we also provide probabilities of improvement in Figure 14. Please refer to the revision for further details.
>
> There are mainly two reasons why we use four random seeds for each experiment on a single scenario. First, the GPU resources we can use are quite limited. Meanwhile, as shown in Table 13, our algorithm has a demand for a relatively high computational cost and computational time. Second, we aim to align with the evaluation protocols used in related works, such as IRIS (single-agent model-based RL), MBVD, and MAZero (multi-agent model-based RL), which also use 4-5 seeds for their experiments. We acknowledge the importance of more random seed to get a rigourous statistical validation, and would be happy to test MARIE for more runs if sufficient computational resources were available.
>
> **Part of W2 & Q2**: The use of SMAC, particularly SMAC v1, is problematic. The evaluation would benefit from using the updated SMAC v2 version.
>
> **Response**: Thanks for the valuable comment. To address the reviewer's concern, we conduct additional experiments on 3 scenarios of SMACv2 -- zerg_5_vs_5, protoss_5_vs_5 and terran_5_vs_5. Without modifying any hyperparameters, MARIE consistently demonstrates higher sample efficiency compared to other baselines in a low data regime. Please refer to Appendix E.2 in the revision for further details.
>
> Regarding the use of SMACv1 in the main experiments, as noted in the SMACv2 paper, certain maps in SMACv1 allow open-loop policies, which only utilize agent IDs and timesteps as input, to achieve non-trivial performance. However, not all maps suffer from this issue. For example, corridor, one of the maps we select, is not affected by this limitation. The significant improvement on the superhard scenario corridor can be a reasonable evidence to demonstrate the effectiveness of our algorithm. Furthermore, our policy is implemented using a 3-layer MLP, only taking the reconstructed observation as input, excluding agent IDs and timesteps. Based on these considerations, we believe that the results of the main experiments on SMACv1 are reliable and effectively validate the superiority of our algorithm.
>
> **Part of W2 & Q3**: A direct comparison on environments beyond SMAC for methods like MAMBA needs to be provided.
>
> **Response**: Thanks for the comment. Upon the reviewer's suggestion, we include MAMBA as a model-based baseline for comparisons on 3 chosen scenarios of MAMujoco. Since MAMBA was originally originally designed for domains with discrete action space, significant effort was required to adapt and evaluate it on MAMujoco, which features continuous action space. Notably, MAMBA fails to enhance policy learning in the Walker2d-v2-2x3 scenario and remains exceptionally time-consuming. Consequently, we report its results only for 1 million environment steps in this scenario. Our MARIE consistently shows superior sample efficiency and achieves the best performance in 2 of 3 scenarios with limited 2M environment steps. For the performance difference between MAMBA and MARIE in HalfCheetah-v2-3x2, we hypothesize that MAMBA's policy learning benefits significantly from using the internal recurrent features of the world model as inputs in this scenario, while the policy in our method only takes the reconstructed observation as input in order to support fast deployment in the environment without the participation of the world model. Please refer to Appendix E.1 in the revision for further details.
>
> **Q4**: Given that the difference in compounding error of the world models between MAMBA and MARIE get worse over time, would the performance gap between the two be reduced if using a smaller imagination horizon for training and does this possibly bridge the gap?
>
> **Response**: Thanks for the question. We report the performance of MARIE and MAMBA on 3s_vs_5z with different imagination horizon. And the result shows that the performance gap between the two is not related to the choice of imagination horizon. Interestingly, a larger imagination horizon may help policy learning in imagination. We hypothesize that longer imagined trajectories help alleviating shortsighted behaviours in policy learning.
>
> | Algorithm | Horizon = 15 | Horizon = 25 |
> | :-: |:-:|:-:|
> | MARIE |0.75±0.09|0.78±0.11|
> | MAMBA |0.13±0.14|0.16±0.13|
>
> ---
> Please do not hesitate to contact us if you need any further clarification or experiments.

---

> > ### Comment · Reviewer_Cbz9 · 2024-11-25
> >
> > Thank you for addressing my initial concerns and providing additional experiments and clarifications. While I appreciate the effort put into responding to my feedback, there are still some unresolved issues that I believe need to be addressed before I can consider revising my assessment.
> >
> > 1. **SMACv2 Comparisons**: The inclusion of SMACv2 experiments is a step in the right direction. However, I notice that MAMBA, the most similar and strongest competing baseline, is absent from these comparisons. Given that MAMBA has been shown to perform comparably in certain environments (as evidenced in MAMujoco), its absence in SMACv2 experiments limits the comprehensiveness of the evaluation. A direct comparison with MAMBA on SMACv2 scenarios would greatly strengthen the claims about MARIE’s superiority.
> >
> > 2. **MAMujoco Results**: The results for MAMBA on MAMujoco—comparable performance in the first HalfCheetah scenario, superior performance in the second, and failure to learn in Walker2d—raise additional questions. It suggests that differences in performance could potentially be attributed to hyperparameter tuning or implementation optimizations rather than fundamental architectural advantages of MARIE. Could you provide more details on the tuning process for MARIE versus MAMBA? Specifically, how much compute and time were allocated to ensure fairness in tuning both methods?
> >
> > 3. **Imagination Horizon Experiment**: While the results for different horizons are appreciated, my original concern was about exploring much shorter horizons for both methods, where compounding error would have minimal time to diverge. This would help clarify whether MARIE’s performance advantages persist under such conditions or if the gap with MAMBA narrows significantly.
> >
> > 4. **Related Work (CoDreamer)**: I still notice no mention of CoDreamer (https://openreview.net/forum?id=f2bgGy7Af7), which introduced a similar concept of decentralized world models. This omission in the related work section continues to detract from the perceived novelty of MARIE. Acknowledging and contrasting with CoDreamer would provide a more complete context for your contributions.
> >
> > 5. **Broader Contributions**: Ultimately, while MARIE introduces some interesting architectural changes, I remain unconvinced that it pushes the boundaries of what is possible in multi-agent world modeling. The results do not present a compelling case that the improvements cannot be achieved by fine-tuning MAMBA or other similar methods. This concern is exacerbated by the lack of rigorous comparisons on all relevant benchmarks and with all appropriate baselines.
> >
> > Thank you again for your efforts to address these points. Whilst I think the work has clearly received a lot of effort and has potential benefit for the community, I do not feel confident that the method truly improves upon prior work. In its current state, I find the results and contributions insufficiently compelling to warrant a change in my recommendation.

---

> ### Author Response · Authors · 2024-11-27
> **Further Response to Reviewer Cbz9 (Part 1/2)**
>
> Thanks for your further feedback. Below, we would like to address your concerns step by step.
>
> **Q1: SMACv2 Comparisons.**
>
> **Response**: Since **MAMBA was originally originally designed for domains with discrete action space and only conducted the main experiment on SMAC**, it would take considerable time and effort to evaluate it on SMACv2. Moreover, due to our limited GPU resources, we were unable to provide MAMBA's experimental results on SMACv2 in our initial response, as we wanted to allow sufficient time for the reviewer to give further feedback. Now, we manage to include the results of MAMBA on SMACv2 and update the figure for plotting the results of all algorithms on SMACv2 in the revision. We refer the reviewer to Figure 11 for details. While MARIE shows competitive performance to MAMBA on zerg_5_vs_5, MARIE is superior to MAMBA in terms of sample efficiency and final performance in the rest 2 scenarios.
>
> **Q2: MAMujoco Results.**
>
> **Response**: First, what we want to emphasize again is that **MAMBA was originally designed for domains with discrete action space and only conducted the main experiment on SMAC**. To ensure that MAMBA could successfully evaluate on the MAMujoco benchmark, **we spent substantial time and effort on its implementation and tuning**.
> 1. Specifically, across all the experiments on MAMujoco, the actor and critic of MAMBA were the same as those of MARIE.
> 2. All parameters, except those related to the difference on world model and corresponding algorithm pipeline, shared between the two methods. For example, the imagination horizon $H$ is set to 15 for both methods, which is also the optimal choice provided in the official repository of MAMBA in the SMAC experiments.
> 3. Furthermore, both approaches were trained on the same server equipped with 1 AMD EPYC 7C13 64-Core Processor and 4 NVIDIA Geforce RTX 4090, ensuring computational fairness.
>
> The averaged training time for a run within 2 million environment steps is reported as follows:
> | Averaged training time  | HalfCheetah-v2-2x3 | HalfCheetah-v2-3x2 | Walker2d-v2-2x3 |
> | :-: |:-:|:-:|:-:|
> | MARIE | 3d 4h | 3d 9h | 5d 20h |
> | MAMBA | 2d 15h | 2d 17h | 2d 20h (until 1 million steps) |
>
> The poor performance of MAMBA on the Walker2d scenario can be attributed to the higher difficulty of this task, which includes a **healthy_reward bonus** (we refer the reviewer to [Walker2d documentation](https://gymnasium.farama.org/environments/mujoco/walker2d/)). This bonus is absent in the HalfCheetah scenario (refer to [HalfCheetah documentation](https://gymnasium.farama.org/environments/mujoco/half_cheetah/) for this notable difference).
>
> MAMBA fails to capture the presence/absence of bonus in the reward function across different scenarios, leading to poor performance of the policy learned in the imaginations of the world model of MAMBA itself. Its performance is even worse than that of model-free algorithms, further highlighting that MAMBA’s world model cannot generalize to complex continuous action space case. Another evidence that support this claim is that during the evaluation of MAMBA, the averaged episode length is 1000 (i.e., the default maximum episode length in MAMujoco) in HalfCheetah without the healthy_reward bonus while it becomes 60~100 in Walker2d. In contrast, the averaged episode length would gradually converge to 1000 across all three scenarios during the evaluation of MARIE.
> We believe that the significant performance gap in the more challenging Walker2d scenario demonstrates that MARIE’s improvements are non-trivial. Additionally, on HalfCheetah, MAMBA exhibits a lack of robustness and consistency, performing worse than MARIE in one case but better in the other.
>
> Considering the overall performance across all three scenarios, especially the pronounced gap in the more difficult Walker2d environment with the healthy_reward bonus, we respectfully disagree with the reviewer’s statement: "Given that MAMBA has been shown to perform comparably in certain environments (as evidenced in MAMujoco) ...".
>
> **Q3: Imagination Horizon Experiment.**
>
> **Response**: We would like to point out that imagination horizon $H = 15$ is the recommended optimal choice provided in the official MAMBA benchmark. We refer the reviewer to [the official link](https://github.com/jbr-ai-labs/mamba/blob/f90d7e5d164c974503dcd97c5cdc5a063cbc90ff/configs/dreamer/optimal/starcraft/LearnerConfig.py#L23) for details.
>
> Additionally, we added the experiment of evaluating MAMBA on 3s_vs_5z with imagination horizon $H = 8$. The comparison between MARIE and MAMBA with different imagination horizon is reported as follows.
> | Method | Horizon = 8 | Horizon = 15 | Horizon = 25 |
> | :-: |:-:|:-:|:-:|
> | MARIE |0.40±0.34|0.75±0.09|0.78±0.11|
> | MAMBA |0.00±0.00|0.13±0.14|0.16±0.13|

---

> > ### Author Response · Authors · 2024-11-27
> > **Further Response to Reviewer Cbz9 (Part 2/2)**
> >
> > **Q4: Related Work (CoDreamer).**
> >
> > **Response**:
> > 1. First, we would like to clarify that the concept of decentralized world models was already introduced in MAMBA [1] which was published in 2022. We do not claim it as our novelty. Instead, we highlight its integration as part of our contributions in the introduction. Specifically, we successfully incorporate the CTDE principle into the design of our **Transformer-based** multi-agent world model, which is quite different from the RSSM-based multi-agent world model like the ones in CoDreamer [2] and MAMBA.
> >
> > 2. Regarding the discussion of CoDreamer, MAMBA extended DreamerV2 [4] within the multi-agent context while CoDreamer was an adaptation of DreamerV3 [5] for the same setting. The primary difference lies in how agent communication is handled: MAMBA employs the self-attention mechanism, whereas CoDreamer uses Graph Neural Networks (GNNs). However, both rely on RSSM (Recurrent State-Space Model) as their foundational world model backbone, which is the core of the Dreamer series [3,4,5]. In contrast, our approach adopts a Transformer-based architecture, casting decentralized dynamics learning as sequence modeling over discrete tokens. Then, the agent-wise aggregation with Perceiver Transformer is a novel solution for computationally efficient agent-wise aggregation in the context of casting decentralized dynamics learning as sequence modeling over discrete tokens. These fundamentally distinguish MARIE from both MAMBA and CoDreamer. Thus, we respectfully disagree with the reviewer’s statement: "This omission in the related work section continues to detract from the perceived novelty of MARIE".
> >
> > 3. Lastly, we find it puzzling that, based on the reviewer's evaluation criteria, CoDreamer itself obviously exhibits significant issues. Specifically, CoDreamer lacks **rigorous evaluation against popular MARL algorithms and appropriate baselines** which the reviewer emphasizes most. In its related work section, it acknowledges both MAMBA and CoDreamer as respective extensions of DreamerV2 and DreamerV3 for multi-agent scenarios, with the primary difference lying in the means for modeling agent-wise communication. However, its experiments only include comparisons against naive baselines, i.e., Independent Dreamer (IDreamer)and Independent PPO (IPPO), while neglecting widely recognized methods like MAPPO, QMIX, and QPLEX, and even the most similar method -- MAMBA.
> >
> > For these reasons, we believe it is not appropriate to include CoDreamer in our related work discussion and comparisons against it.
> >
> > **Reference**
> >
> > [1] Egorov, Vladimir, and Aleksei Shpilman. "Scalable multi-agent model-based reinforcement learning." arXiv preprint arXiv:2205.15023 (2022).
> >
> > [2] Toledo, Edan, and Amanda Prorok. "CoDreamer: Communication-Based Decentralised World Models." arXiv preprint arXiv:2406.13600 (2024).
> >
> > [3] Hafner, Danijar, et al. "Dream to control: Learning behaviors by latent imagination." arXiv preprint arXiv:1912.01603 (2019).
> >
> > [4] Hafner, Danijar, et al. "Mastering atari with discrete world models." arXiv preprint arXiv:2010.02193 (2020).
> >
> > [5] Hafner, Danijar, et al. "Mastering diverse domains through world models." arXiv preprint arXiv:2301.04104 (2023).
> >
> > **Q5: Broader Contributions.**
> >
> > **Response**: We would like to respond point by point to the concern -- "the lack of rigorous comparisons on all relevant benchmarks and with all appropriate baselines" raised by the reviewer.
> > * **Rigorous comparisons**: We **followed the reviewer's suggestions by using [RLiable](https://github.com/google-research/rliable) to evaluate the main experiments on SMAC**, and plotted the results with statistical metrics such as Median, IQM, Mean, and Optimality Gap, shown as Figures 13 and 14 in Appendix F. The results were also explicitly mentioned in our previous response to "W1 & Q1". Across all these metrics, including the probability of improvement, MARIE consistently and significantly outperforms all baselines, delivering a statistically rigorous validation. **We kindly request that  reviewer Cbz9 carefully review the content in Appendix F**.
> > * **Relevant benchmarks**: We have conducted experiments on a broad range of benchmarks, including SMAC, SMACv2, and MAMujoco, covering both discrete action space case and continuous action space case. We believe that these diverse benchmarks adequately demonstrate the improvements of MARIE.
> > * **Appropriate baselines**: We have chosen popular model-free baselines and strong model-based baselines, 5 baselines in total for comparisons, as outlined in the experimental setup section. The selection ranging from model-free algorithms to model-based algorithms, provides a solid foundation for meaningful and comprehensive comparisons.

---

> > > ### Comment · Reviewer_Cbz9 · 2024-11-27
> > >
> > > Thank you for providing the additional experiments. Evaluating MAMBA on SMACv2 again shows comparable performance on 2 out of 3 tasks. I describe the performance as comparable because with only 4 seeds, the high variance inherent in RL and these environments significantly influences the results. Given this, and upon reviewing the curves for both SMACv2 and MAMujoco, I find it difficult to confidently assert that MARIE demonstrates superior performance.
> > >
> > > Regarding the additional horizon experiments, I believe it would be valuable to include these in the paper to address potential questions from future readers.
> > >
> > > As for your final point, I am not specifically reviewing MAMBA or CoDreamer, both of which also exhibit notable issues. My comment was focused on how their existence impacts the perceived novelty of this work. A transformer is a graph neural network with a fully connected adjacency matrix and additional enhancements. Claiming that the use of a transformer sufficiently distinguishes your approach to omit reference to such methods in the related work section is, in my view, inaccurate.
> > >
> > > That said, most of my concerns have been addressed, and I appreciate your efforts in this regard. I will adjust my score upward, as the use of this architecture in the proposed manner is interesting and could contribute to the community. However, I remain uncertain whether MARIE represents a significant enough step forward to be truly impactful.

---

> > > > ### Author Response · Authors · 2024-11-28
> > > > **Thanks for Reviewer Cbz9 and hope for earning your higher score**
> > > >
> > > > Thank you for your thoughtful and constructive feedback. We are really happy to see that our efforts and dedication to address your concerns are recognized and valued. Here, we would like to express our heartfelt gratitude for your decision to adjust the score upward.
> > > >
> > > > We learn an important and valuable lesson from the reviewer about the necessity of rigorous statistical validation. **To reflect this, we explicitly highlight the limitations of our current experiments in the *Conclusion and Limitation* section of the latest revision**. We acknowledge that with four limited seeds, the high variance inherent in RL and these environments impacts the confidence of performance evaluations, and we appreciate your comment on this point and your suggestion on using standard protocol such as RLiable.
> > > >
> > > > Upon on your suggestion, **we also incorporate the additional horizon experiments into Appendix E.5 to address potential questions from future readers**. We believe the additions enhances the comprehensiveness and utility of the paper.
> > > >
> > > > Since we are not familiar with the topic of GNNs, we apologize for the misunderstanding of your comment regarding the reference to CoDreamer in our previous response. Now we better understand your perspective and agree with you that a transformer with bi-directional attention mechanism can be viewed as a special case of graph neural network with a fully connected adjacency matrix. To this end, it is reasonable and also necessary to clarify the difference between the aggregation of MARIE and CoDreamer.
> > > > We recognize that CoDreamer, which uses GAT V2 for communication, share some conceptual and architectural similarity with our use of Perceiver for aggregation. However, our primary focus lies in computational efficiency, whereas CoDreamer emphasizes the underlying topological graph structure among agents —- a perspective that MARIE does not currently account for. **We now include a discussion of this distinction in an independent section in appendix (Refer to Appendix F for details)**. Furthermore, we believe its integration would be a promising direction for future improvements to MARIE.
> > > >
> > > > Lastly, we would like to address your remaining concern about whether MARIE represents a significant enough step forward to be truly impactful. Could you please clarify what specific issues remain, and how we might address them? We are dedicated to further improving our work based on your valuable insights, in the hope of earning your further recognition and potentially a higher score.
> > > >
> > > > Thank you once again for your valuable feedback and support throughout the review process.

---

### Official Review · Reviewer_qbLx · 2024-11-04

**Soundness:** 3
**Presentation:** 3
**Contribution:** 3
**Rating:** 6
**Confidence:** 2

**Summary:**

The authors propose Perceiver Transformer-based world model for MARL that addresses the actual problems of scalalability and the non-stationarity. The evaluations on SMAC and MAMujuco prove the model superiority over multiple baselines.

**Strengths:**

- The proposed model combined decentralized dynamics modeling with centralized representation aggregation using Transformer sequence modeling.
- The paper is well-written and easy to follow.
- The authors provide the ablation results and the analysis of attention patterns to reveal the implicit decison-making features.

**Weaknesses:**

- The paper presentation could be improved with captioning figures of experimental results with short conclusions

**Questions:**

- The paper presentation would benefit from increasing the font size of figures in the main text.

**Details Of Ethics Concerns:**

-

---

> ### Author Response · Authors · 2024-11-22
> **Response to Reviewer qbLx**
>
> Dear Reviewer qbLx,
>
> We sincerely appreciate your precious time and constructive comments. In the following, we would like to answer your concerns separately.
>
> **W1**: The paper presentation could be improved with captioning figures of experimental results with short conclusions
>
> **Response**: Thanks for the valuable comment. We have added a corresponding conclusion in Figure 3~6. Please see the revision for details.
>
> **Q1**: The paper presentation would benefit from increasing the font size of figures in the main text.
>
> **Response**: Thanks for the valuable comment. In response to your suggestion, we have redrawn Figure 1~2 in the main text and increased the font size to enhance readability.
>
> Please do not hesitate to contact us if you need any further clarification or experiments.

---

> > ### Comment · Reviewer_qbLx · 2024-11-29
> >
> > Thank you for addressing my comments. The updated figures improve the paper readability and make the results more clear.

---

> ### Author Response · Authors · 2024-11-29
> **Thanks for Reviewer qbLx's feedback.**
>
> Dear Reviewer qbLx,
>
> Thank you for your continued feedback. We are grateful that our efforts have addressed your concerns.
>
> We believe that your comments have notably improved the quality of our paper. Additionally, we would like to bring your attention to new experiments conducted during the discussion period. Following the insightful suggestions of Reviewer ysTM, we have conducted more comprehensive experiments, including an analysis of how errors in reconstructed observations impact final performance, an evaluation of the accuracy of discount prediction and the final performance with an increasing imagination horizon, as well as a comparison between different aggregation methods. Furthermore, thanks to the valuable feedback from Reviewer Cbz9, we have provided a statistically rigorous validation of our main experiment on SMAC and evaluated MARIE on an additional benchmark, SMACv2, to further test MARIE's superiority. As a result of these revisions, among other improvements, Reviewer ysTM and Cbz9 increased their scores from 5 to 6 and from 3 to 5, respectively.
>
> If you have any remaining questions during the discussion period, we would be happy to answer them. And we are dedicated to earning a higher score from you.

---

### Official Review · Reviewer_ysTM · 2024-11-04

**Soundness:** 2
**Presentation:** 3
**Contribution:** 2
**Rating:** 6
**Confidence:** 4

**Summary:**

This paper introduces a novel framework, **MARIE (Multi-Agent auto-Regressive Imagination for Efficient learning)**, which leverages a Transformer-based world model to enable sample-efficient policy learning in **Multi-Agent Reinforcement Learning (MARL)**. MARIE addresses two key challenges in MARL: scalability and **non-stationarity**. The approach combines decentralized local dynamics learning with centralized feature aggregation using a Perceiver Transformer. The authors evaluate the proposed method on the Starcraft Multi-Agent Challenge (SMAC) and **MAMuJoCo**, demonstrating improved sample efficiency and performance over existing model-free and model-based methods.

**Strengths:**

1. The integration of decentralized local dynamics learning and centralized feature aggregation is well-motivated and effectively addresses key challenges in MARL, such as scalability and non-stationarity.
2. The use of the Perceiver Transformer for centralized representation aggregation is an innovative contribution that facilitates efficient global information sharing between agents while maintaining scalability.

**Weaknesses:**

1. **Necessity of individual components**: The authors claim that this work is “the first pioneering Transformer-based world model for multi-agent systems,” but the underlying techniques—centralized feature aggregation, the Perceiver Transformer, and autoregressive modeling of discretized tokens—are already present in the literature. More ablation experiments to demonstrate the necessity of these components would strengthen the paper. It is necessary to investigate whether it is a kind of simple combinations of different techniques, or more reasonable design for MARL.
2. **Limited comparison to existing Transformer-based world models**: While the paper compares its method with model-free and some model-based MARL approaches, a more in-depth exploration of existing Transformer-based methods in MARL, or related architectures from single-agent RL that could directly be extended to MARL (e.g., IRIS, TWM and other methods), is lacking. It would be beneficial to further discuss why existing single-agent Transformer-based approaches cannot be directly adapted to MARL.

**Questions:**

- Given that the global state is known, could it be directly used as the aggregated global feature? If analysis or experiments were performed to validate the effectiveness of the current agent-wise aggregation, it would be more convincing.
- Since the policy relies on reconstructed observations, a deeper analysis of how errors in reconstructed observations impact final performance would be insightful.
- "The policies π are exclusively trained using imagined trajectories." Does this lead to wasted real experience collected during training?
- I am curious about the prediction accuracy for discounts at each step. As the horizon (H) increases, can the model accurately predict the end of the game, and how does this affect performance?
- Since MARIE separates model learning from policy learning, providing intuitive or experimental comparisons with methods that jointly learn the model and policy would increase the persuasiveness of the approach. For example, the following references could be useful:
  - [1] Benjamin Eysenbach, Alexander Khazatsky, Sergey Levine, and Ruslan Salakhutdinov. Mismatched no more: Joint model-policy optimization for model-based RL. Advances in Neural Information Processing Systems, 35:23230–23243, 2022.
  - [2] Raj Ghugare, Homanga Bharadhwaj, Benjamin Eysenbach, Sergey Levine, and Ruslan Salakhutdinov. Simplifying model-based RL: learning representations, latent-space models, and policies with one objective. arXiv preprint arXiv:2209.08466, 2022.
- In the first ablation experiment, regarding learning local dynamics instead of joint dynamics, the authors state that “the scalability issue is exacerbated by a growing number of agents.” However, in Figure 4, the performance of learning joint dynamics degrades more with 3 agents (3s_vs_3z) than with 5 agents (2s3z). This seems inconsistent with the authors' claim.
- In the third ablation study, it would be worth exploring the effect of not discretizing observations and instead learning a continuous embedding through a linear layer. Additionally, assessing the impact of policies that directly depend on the internal hidden states of the Transformer, rather than reconstructed observations, would be insightful. Intuitively, the policy input only needs to capture decision-relevant information, not a complete image reconstruction. Moreover, errors in reconstruction may negatively impact policy learning.
- In Figure 7, the authors claim that MARIE “has remarkably better error,” but the actual curves do not seem to support the term “remarkably.” Providing a corresponding performance comparison curve would make this claim more visually intuitive.
- Including a more comprehensive set of experimental results in Table 1 would enhance the paper.
- Can the authors provide more details on the computational cost of using a Perceiver Transformer for centralized aggregation? How does this affect MARIE's scalability as the number of agents increases?
- Could the authors clarify the role of intra-step autoregression and how it contributes to the overall performance of the model? A comparison between the Perceiver and other common aggregation methods would also be helpful.
- The paper shows strong performance on simulation benchmarks, but adding results or discussions on how MARIE could be applied to real-world scenarios would increase its impact and relevance.
- The Preliminary section could benefit from a brief introduction to the Perceiver and other aggregation techniques, making the paper more accessible to readers unfamiliar with these concepts.
- Are there any limitations of MARIE that might make it less effective in certain multi-agent settings, such as environments with highly heterogeneous agents or asymmetric observation spaces?

---

> ### Author Response · Authors · 2024-11-22
> **Response to Reviewer ysTM (Part 1/4)**
>
> Dear Reviewer ysTM,
>
> We sincerely appreciate your precious time and constructive comments. In the following, we would like to answer your concerns separately.
>
> **W1**: Necessity of individual components.
>
> **Response**: Thanks for the constructive comment.
> * First, we would like to clarify that our work is pioneering in the sense that it introduces the first effective transformer-based world model within the multi-agent context, enabling multi-agent world models to enjoy the benefit of recent progress on powerful generative models.
> * Second, **casting local dynamics learning as autoregressive modeling over discretized tokens is inspired by IRIS[1]**. Compared with TWM[2] which used autoregressive modeling over continuous embedding, IRIS attained superior performance to TWM on Atari 100K, which demonstrates that autoregressive modeling over discrete tokens can further unleash the power of Transformer on dynamics learning. It motivates us to adopt a similar manner to build the Transformer-based multi-agent world model.
> * For **the Perceiver Transforme**, we conduct additional experiment to compare our proposed aggregation via Perceiver with the common aggregation with self-attention. The difference on the final performance and the FLOPs shows that our agent-wise aggregation with Perceiver offers a more computationally efficient solution compared to self-attention aggregation. **See our response to Q10 for details**.
> * For **centralized feature aggregation**, we show that the world model struggles to capture the underlying multi-agent dynamics in the trajectories and further hinders the policy learning without the agent-wise information aggregation in the second ablation experiment.
>
> **W2**: Limited comparison to existing Transformer-based world models.
>
> **Response**: Thank you for the valuable comment. Existing Transformer-based world models are primarily designed for single-agent scenarios, but they can be naturally adapted to multi-agent settings, modeling either independently local dynamics or joint dynamics. Based on the reviewer's suggestion, we have included IRIS as a Transformer-based world model baseline. Notably, the **"Centralized Manner"** and **"MARIE w/o aggregation"** variants from our ablation experiments correspond to IRIS baseline variants under different deployment strategies. Here we would like to clarify that these IRIS baseline variants also uses the same actor-critic method as MARIE during learning in imaginations phase, which can form a more fair comparison. To deliver a more intuitive demonstration, we additionally plot a new figure in Appendix E.4. Shown by the result, without incorporating CTDE principle, the learning of single-agent world model would be disrupted by the scalability and non-stationarity issues, validating our motivation in the introduction.
>
> **Q1**: Given that the global state is known, could it be directly used as the aggregated global feature? If analysis or experiments were performed to validate the effectiveness of the current agent-wise aggregation, it would be more convincing.
>
> **Response**: Thanks for the insightful question. In real-world applications, access to the global state, as available in simulations, is often impractical. Therefore, we opted not to incorporate the global state into the training of the world model, including using it directly as the aggregated global feature.
> Furthermore, incorporating the global state during training would necessitate its estimation during inference. Without global state estimation, long-term continuous imagination within the world model would require querying the simulation environment for intermediate global states, which is infeasible. Additionally, estimating the global state would introduce more prediction errors.
>
> **Q2**: Since the policy relies on reconstructed observations, a deeper analysis of how errors in reconstructed observations impact final performance would be insightful.
>
> **Response**: Thanks for the question. The policy uses reconstructed observations as input to avoid a input distribution shift between collecting experience from real environments and learning in imagination. Here, we conduct additional experiment with setting the number of observation token to 8 which results in more errors in reconstructed observations. We report the result within 50 thousand (50000) environment steps in the following table:
>
> | Map    | MARIE (16 tokens) | MARIE (8 tokens) |
> | :-: |:-:|:-:|
> | 2m_vs_1z |0.96±0.07|0.74±0.42|
>
> As the reconstructed observation contains more errors, the performance of the policy learned in imaginations also significantly degrades.

---

> > ### Author Response · Authors · 2024-11-22
> > **Response to Reviewer ysTM (Part 2/4)**
> >
> > **Q3**: "The policies $\pi$ are exclusively trained using imagined trajectories." Does this lead to wasted real experience collected during training?
> >
> > **Response**: Thank you for the question. This does not lead to wasted real experience, as all collected trajectories are fully utilized for training the world model. The world model, in turn, generates imagined trajectories that are used to train the policy. This process ensures that the real experience contributes indirectly to policy optimization by improving the quality and accuracy of the world model.
> >
> > **Q4**: I am curious about the prediction accuracy for discounts at each step. As the horizon (H) increases, can the model accurately predict the end of the game, and how does this affect performance?
> >
> > **Response**: Thanks for the question. We first trained a world model with an imagination horizon of 25 and used the resulting policy to sample 20 trajectories in the 3s_vs_5z environment. Given a specified imagination horizon, we randomly cropped trajectory segments and generated imagination solely within this world model by autoregressively rolling out action sequences. Since the predicted discount in our world model follows a Bernoulli likelihood, which softly accounts for the possibility of episode termination, we convert the predicted discount to 0 or 1 based on whether it exceeds 0.5. We then compare these predictions against the actual episode terminations to compute the accuracy. The results are presented as follows.
> >
> > |     | Horizon = 5 | Horizon = 8 | Horizon = 10 | Horizon = 15|Horizon = 20|Horizon = 25|
> > | :-: |:-:|:-:|:-:|:-:|:-:|:-:|
> > | Predicted Discount Accuracy |0.997±0.024|0.996±0.032|0.996±0.025|0.999±0.006|0.992±0.037|0.998±0.017|
> >
> > Additionally, we evaluate the performance of MARIE with different imagination horizon.
> > | Map    | Horizon = 8 | Horizon = 15 | Horizon = 25 |
> > | :-: |:-:|:-:|:-:|
> > | 3s_vs_5z |0.40±0.34|0.75±0.09|0.78±0.11|
> >
> > **Q5**: Since MARIE separates model learning from policy learning, providing intuitive or experimental comparisons with methods that jointly learn the model and policy would increase the persuasiveness of the approach.
> >
> > **Response**: Thanks for the suggestion. Our approach follows the **learning in imaginations** paradigm adopted by methods like DreamerV1 [3], IRIS [1], and TWM [2]. The successes of these methods, along with ours, confirm the effectiveness of this paradigm in model-based reinforcement learning (MBRL). Also, we have compared MARIE with a baseline that jointly learns the model and policy (i.e., MBVD [4]). The results demonstrate that policy learning benefits significantly from the accurate and consistent imagination from the separately trained world model, underscoring the advantages of our approach.
> >
> > **Q6**: In the first ablation experiment, regarding learning local dynamics instead of joint dynamics, the authors state that “the scalability issue is exacerbated by a growing number of agents.” However, in Figure 4, the performance of learning joint dynamics degrades more with 3 agents (3s_vs_3z) than with 5 agents (2s3z). This seems inconsistent with the authors' claim.
> >
> > **Response**: Thanks for the question. Here, we would like to explain the reason why the performance of learning joint dynamics degrades more with 3 agents (3s_vs_3z) than with 5 agents (2s3z). Although 2s3z involves more agents, its overall task complexity is lower compared to 3s_vs_3z. As shown in Table 1, model-free methods achieve better performance in 2s3z than in 3s_vs_3z, highlighting the greater difficulty of 3s_vs_3z. It means that it needs to take more interactions to learn an optimal policy in 3s_vs_3z. When the world model fails to capture the dynamics accurately, the imagined trajectories deviate substantially from the real environment, leading to degraded policy performance which is particularly amplified by higher difficulty in 3s_vs_3z. The observed degradation seems inconsistent but remains reasonably aligned with our claim.
> >
> > **Q7**: In the third ablation study, it would be worth exploring the effect of not discretizing observations and instead learning a continuous embedding through a linear layer. Additionally, assessing the impact of policies that directly depend on the internal hidden states of the Transformer, rather than reconstructed observations, would be insightful. Intuitively, the policy input only needs to capture decision-relevant information, not a complete image reconstruction. Moreover, errors in reconstruction may negatively impact policy learning.

---

> > > ### Author Response · Authors · 2024-11-22
> > > **Response to Reviewer ysTM (Part 3/4)**
> > >
> > > **Response**: Thanks for the thoughtful question. Using the internal hidden states of the Transformer-based world model as inputs could indeed benefit policy learning by providing rich representations maybe relevant to decision-making. However, this approach would necessitate retaining the world model during decentralized execution to compute these hidden states, which contradicts the primary goal of the world model: to accelerate multi-agent policy learning while ensuring lightweight and efficient policy inference during deployment. Additionally, the autoregressive nature of Transformers would significantly increase computational overhead, making this approach less practical for real-time deployment. These trade-offs suggest that the current design, which relies on reconstructed observations, strikes a better balance between performance and efficiency.
> > >
> > > **Q8**: In Figure 7, the authors claim that MARIE “has remarkably better error,” but the actual curves do not seem to support the term “remarkably.” Providing a corresponding performance comparison curve would make this claim more visually intuitive.
> > >
> > > **Response**: Thanks for the comment. The claim of “remarkably better error” is supported by the cumulative error curves in Figure 7, which show MARIE maintaining errors below 2 across all agents at a long imagination horizon (H = 25), while MAMBA's cumulative error exceeds 2. To improve visualization clarity, we applied clipping to the y-axis at an error level of 2. Although both MARIE and MAMBA exhibit exponential growth in cumulative error, MARIE demonstrates significantly better scalability at longer horizons, as its error remains bounded within this range, unlike MAMBA.
> > >
> > > To further validate this claim, we provide a quantitative comparison of final performance under an imagination horizon of 25:
> > > | Map    | MARIE (Horizon = 25) | MAMBA (Horizon = 25) |
> > > | :-: |:-:|:-:|
> > > | 3s_vs_5z |0.78±0.11|0.16±0.13|
> > >
> > > These results clearly illustrate MARIE's superior performance under long-horizon settings, both in terms of cumulative error and final task performance.
> > >
> > > **Q9**: Including a more comprehensive set of experimental results in Table 1 would enhance the paper.
> > >
> > > **Response**: Thanks for the suggestion. We believe the set of experiments already presented in Table 1 is comprehensive. As stated in the experimental setup, we selected 13 representative scenarios from SMAC, covering three levels of difficulty: Easy, Hard, and SuperHard. In contrast, other Multi-Agent Model-based RL methods, e.g., MAZero[5] and MBVD[4], usually choose 8~9 scenarios for evaluation. A thorough comparison among MARIE and all other baselines across all these chosen scenarios has been provided in Table 1. If provided with enough GPU devices, we would be happy to evaluate MARIE on all available scenarios in SMAC.
> > >
> > > **Q10**: Can the authors provide more details on the computational cost of using a Perceiver Transformer for centralized aggregation? How does this affect MARIE's scalability as the number of agents increases?
> > >
> > > **Response**: Thanks for the question. We report the FLOPs for both Perceiver and Self-Attention aggregation with varying number of agents, as shown in the table below. The Perceiver aggregation in our algorithm consists of one transformer layer with cross-attention and two transformer layers with self-attention. To form a fair comparison, the self-attention aggregation also consists of three transformer layers with self-attention.
> > >
> > > |     | 2 agents | 3 agents | 5 agents | 9 agents |
> > > | :-: |:-:|:-:|:-:|:-:|
> > > | Perceiver Aggregation |0.016GFLOPs|0.024GFLOPs|0.041GFLOPs|0.073GFLOPs|
> > > | Self-Attention Aggregation |0.133GFLOPs|0.201GFLOPs|0.335GFLOPs|0.603GFLOPs|
> > >
> > > Additionally, we compare the performance of these aggregation methods in the 2m_vs_1z scenario after 50000 steps:
> > >
> > > | Map    | Perceiver Aggregation (3 layers) | Self-Attention Aggregation (3 layers) |
> > > | :-: |:-:|:-:|
> > > | 2m_vs_1z |0.96±0.07|0.54±0.46|
> > >
> > > These results show that Perceiver aggregation offers a more computationally efficient solution compared to self-attention aggregation.
> > >
> > > **Q11**: Could the authors clarify the role of intra-step autoregression and how it contributes to the overall performance of the model? A comparison between the Perceiver and other common aggregation methods would also be helpful.
> > >
> > > **Response**: Thanks for the question. First, the meaning of intra-step autoregression is clearly stated in Line 219. Second, a comparison between the Perceiver and the common self-attention aggregation is provided in **the response to Q10**. Please see it for details.

---

> > > > ### Author Response · Authors · 2024-11-22
> > > > **Response to Reviewer ysTM (Part 4/4)**
> > > >
> > > > **Q12**: The paper shows strong performance on simulation benchmarks, but adding results or discussions on how MARIE could be applied to real-world scenarios would increase its impact and relevance.
> > > >
> > > > **Response**: Thanks for the suggestion. At this stage, we are not considering real-world experiments due to limitations in hardware and real-world data. However, we acknowledge the importance of bridging the gap between simulation and real-world applications, and we hope to explore this direction in future work when everything is ready.
> > > >
> > > > **Q13**: The Preliminary section could benefit from a brief introduction to the Perceiver and other aggregation techniques, making the paper more accessible to readers unfamiliar with these concepts.
> > > >
> > > > **Response**: Thanks for the suggestion. Due to page limitations, we are unable to include a detailed introduction to the Perceiver and other aggregation techniques in Preliminary section of this version. However, we would address this if this paper is accepted, to make the paper more accessible to readers unfamiliar with these concepts.
> > > >
> > > > **Q14**: Are there any limitations of MARIE that might make it less effective in certain multi-agent settings, such as environments with highly heterogeneous agents or asymmetric observation spaces?
> > > >
> > > > **Response**: Thanks for the question. In environments with highly heterogeneous agents, sharing local dynamics could pose challenges, as the dynamics might differ significantly between agents. This could be a limitation of MARIE in such settings, where the assumptions of shared dynamics may not hold across all agents.
> > > >
> > > > **Reference**
> > > >
> > > > [1] Micheli, Vincent, Eloi Alonso, and François Fleuret. "Transformers are Sample-Efficient World Models." The Eleventh International Conference on Learning Representations, 2023.
> > > >
> > > > [2] Robine, Jan, et al. "Transformer-based World Models Are Happy With 100k Interactions." The Eleventh International Conference on Learning Representations, 2023.
> > > >
> > > > [3] Hafner, Danijar, et al. "Dream to control: Learning behaviors by latent imagination." arXiv preprint arXiv:1912.01603 (2019).
> > > >
> > > > [4] Xu, Zhiwei, et al. "Mingling foresight with imagination: Model-based cooperative multi-agent reinforcement learning." Advances in Neural Information Processing Systems 35 (2022): 11327-11340.
> > > >
> > > > [5] Liu, Qihan, et al. "Efficient Multi-agent Reinforcement Learning by Planning." arXiv preprint arXiv:2405.11778 (2024).
> > > >
> > > > ---
> > > > **At the very end**, given the extensive efforts we have invested in addressing total 16 questions reviewer ysTM raises, **we kindly request that reviewer ysTM carefully review our detailed responses**.

---

> > > ### Comment · Reviewer_ysTM · 2024-11-27
> > > **Discussion (Part2-4)**
> > >
> > > **Q3 & Q5:**
> > > The current training pipeline design in this paper appears to be rational. Unlike classical MBPO in model-based RL, which integrates real and imagined data for policy training, this approach employs real data to train the world model and subsequently uses the world model to generate data for policy training separately. However, the transition between these two training phases may necessitate adjustments to certain hyperparameters, particularly to ensure that the world model adequately captures the real data. Additionally, from the perspective of online RL, environments requiring continuous exploration may necessitate alternating between world model and policy training.
> > >
> > > **Q4:**
> > > The authors' experiments validate the effectiveness of discount prediction within the range of H within [5, 25], demonstrating strong performance in the SMAC environment. While this may exhibit different characteristics in other RL environments, it is not the primary focus of this paper.
> > >
> > > **Q6:**
> > > I understand the inherent complexity differences between the two maps. To better illustrate the impact of increasing agent numbers, I suggest controlling the variables in the comparison. Since the SMAC environment allows for custom maps, you could fix the unit types and design maps with varying agent counts for comparison. This paper should include similar experiments; you might refer to "Li, Chuming, et al., "Ace: Cooperative multi-agent q-learning with bidirectional action-dependency." Proceedings of the AAAI conference on artificial intelligence. Vol. 37. No. 7. 2023." for inspiration.
> > >
> > > **Q7:**
> > > I appreciate the authors' trade-off regarding real-time performance, acknowledging that the current design is a balanced solution. Future work could explore efficient decentralized deployment of multi-agent transformers, aiming for more impressive efficiency in scenarios with a large number of agents.
> > >
> > > **Q8:**
> > > The final performance is indeed more convincing. Regarding sequence prediction comparisons, compounding error may not be a significant metric. The authors might consider alternative analytical indicators, such as bi-simulation.
> > >
> > > **Q9:**
> > > Previous studies have provided tabular performance for all SMAC maps, which is a common knowledge in the MARL cooperation community. However, the differences between SMAC maps are relatively minor, and recent model-free methods have achieved very high performance. Therefore, I suggest that future work consider more complex MARL environments, with SMAC serving as an example validation.
> > >
> > > **Q10 & Q11:**
> > > The effectiveness of Perceiver Aggregation is indeed commendable.
> > >
> > > **Q13 & Q14:**
> > > In future iterations, consider adding more detailed introductions and discussions on limitations in the appendix. Additionally, open-sourcing the code would significantly enhance the paper's impact and the subsequent work it inspires.
> > >
> > > Lastly, I would like to express my gratitude to the authors for their patient rebuttal. I hope that these comments and suggestions will contribute to the further enhancement of this research topic.

---

> > ### Comment · Reviewer_ysTM · 2024-11-24
> > **Discussion**
> >
> > Thank you for your detailed response and the additional explanations and experiments provided. I believe these enhancements address some parts of my concerns. However, I would suggest that the authors delve deeper into the unique aspects of transitioning from single-agent to multi-agent transformer world-models, as this could be a significant potential contribution of your work. While the CTDE paradigm has been a notable approach in multi-agent reinforcement learning, focusing solely on simulation environments (with relatively simple coordination and fewer agents, <50) may not fully demonstrate the strengths of this paradigm. Therefore, I recommend that the authors not limit themselves to CTDE and explore the scaling-up capabilities of transformer architectures in MARL, which could be a more valuable topic of discussion.
> >
> > Regarding Q1, I find the response somewhat evasive. Since your experiments are conducted in simulation environments, the mention of global state issues in real-world applications seems to shift the focus. Nonetheless, this does not detract from your contributions. Future research could consider leveraging global state more effectively to push the performance of agents to a higher level.
> >
> > I have decided to increase my review score to 6, reflecting my improved assessment of the work. I will provide further feedback at a later time to assist the authors in refining their work.

---

> > > ### Author Response · Authors · 2024-11-25
> > > **Thanks for Reviewer ysTM's Response**
> > >
> > > Dear Reviewer ysTM,
> > >
> > > We would like to express our heartfelt gratitude for your positive response and detailed evaluation. Your constructive suggestions and decision to increase the score are deeply appreciated.
> > >
> > > Regarding the unique aspects of transitioning from single-agent to multi-agent transformer world-models, we fully agree with the reviewer that there are more potential approaches (e.g., the architecture design within the multi-agent context) not limited to incorporating CTDE. Your suggestion to explore the scaling capabilities of transformer architectures in multi-agent reinforcement learning (MARL)—especially beyond the CTDE paradigm and into larger-scale coordination scenarios—is highly insightful. While our current work primarily focuses on the CTDE framework within simulation environments, scaling up is indeed essential to more deeply investigate the potential of transformer-based architectures in MARL. Future work would aim at exploring these scaling capabilities in more complex multi-agent systems.
> > >
> > > Regarding Q1, we sincerely apologize if our initial response came across as evasive. This was not our intention. Rather, we aimed to highlight that our design choices were motivated by potential real-world applications, where reliance on a global state may not always be feasible. In other words, it was a cautious consideration for any potential opportunity to seamlessly apply it to the real-world scenarios. But we are in favour of the reviewer's comment on leveraging the global state in the world model. This is a valuable point that warrants further investigation, and we would clarify this in the revised manuscript.
> > >
> > > Once again, we are grateful for your constructive feedback and valuable suggestions. We are committed to refining our work based on your insights and look forward to any further comments you may provide to help us improve.

---

### Official Review · Reviewer_GJdy · 2024-11-04

**Soundness:** 2
**Presentation:** 3
**Contribution:** 2
**Rating:** 6
**Confidence:** 3

**Summary:**

Considering the inevitable challenges of both centralized and decentralized learning in developing a world model, this paper proposes MARIE (Multi-Agent auto-Regressive Imagination for Efficient learning), a Transformer-based approach that integrates both methods. The process is divided into three stages: the first involves collecting multi-agent trajectories, the second focuses on learning the world model from these experiences, and the third uses the world model for policy learning. The second stage, which centers on the learning of the world model, involves discretizing observations with the VQ-VAE method and using the learned discretized codes to construct a Transformer architecture for transition prediction. Additionally, the authors incorporate agent-wise aggregation information to mitigate non-stationarity. Experiments on SMAC and MAMujoco are conducted to validate the method's effectiveness.

**Strengths:**

1.	In constructing the world model, the authors considered both centralized information and decentralized information.
2.	The overall logic of the paper is coherent and easy to understand.
3.	The paper conducted extensive experiments.

**Weaknesses:**

1.	The learning results of the world model depend on the supervisory signals, specifically the trajectories generated by a superior policy used as labels. In complex scenarios, without trajectories produced by an optimal policy, it may be difficult to learn a complete dynamic transition.

**Questions:**

1.	Considering that in SMAC all agents share the same environment reward, is the $r_t^i$ predicted by the "Shared Transformer" the same for each agent? If they are different, how is the team reward used to train the agents during the imagination phase? Is it averaged from $\{r_t^i \}_{i=1}^{N}$?
2.	Based on Question 1, did the authors consider having each agent learn a different reward while learning the world model, in order to address the credit assignment problem in MARL through the world model learning process?
3.	In the training process of the world model, how are the trajectories used as labels obtained? Also, please discuss what should be done in complex scenarios when there is no good initial policy to generate the trajectories.
4.	Please explain in detail the role of learning $\gamma$ in the overall method, and why it cannot be replaced with a constant.
5.	How is the codebook $Z$ initialized, and does the initialization affect the learning outcomes under different conditions?

---

> ### Author Response · Authors · 2024-11-22
> **Response to Reviewer GJdy (Part 1/2)**
>
> Dear Reviewer GJdy,
>
> We sincerely appreciate your precious time and constructive comments. In the following, we would like to answer your concerns separately.
>
> **W1**: The learning results of the world model depend on the supervisory signals, specifically the trajectories generated by a superior policy used as labels. In complex scenarios, without trajectories produced by an optimal policy, it may be difficult to learn a complete dynamic transition.
>
> **Response**: Thanks for the comment. While it is possible to train a world model from a pre-collected offline dataset, our algorithm adopts a Dreamer[1]-like scheme (i.e., learning in imaginations) to train and leverage the world model, which runs the following three parts iteratively:
> 1. collect experience by executing the policy;
> 2. learn the world model from the collected experience;
> 3. learning the policy with the rollouts from the world model.
>
> Thus, our algorithm doesn't need a superior policy to generate trajectories for world model training since the learning of world model is coupled with a randomly initialized policy online learning inside it. If the world model can provides trajectories that are sufficiently accurate compared to reality, the policy can progressively converge to an optimal one.
>
> **Q1**: Considering that in SMAC all agents share the same environment reward, is the $r_t^i$ predicted by the "Shared Transformer" the same for each agent? If they are different, how is the team reward used to train the agents during the imagination phase? Is it averaged from $\sum_{i=1}^Nr_t^i/N$?
>
> **Response**: Thanks for the question. Yes, the label of predicting $r_t^i$ for each agent is the same, i.e., the ground-truth team reward in the SMAC. Here, we suppose that using the team reward as a shared label for each agent's reward prediction can be considered as a simple yet potentially inappropriate credit assignment where each agent makes equal contribution and gets their individual reward $r_{\text{team}} / N$ which is further magnified by a factor of constant $N$. Hence, we use the individual predicted reward $\hat{r}_{t}^i$ instead of the averaged reward when training each agent during the imagination phase.
>
> **Q2**: Based on Question 1, did the authors consider having each agent learn a different reward while learning the world model, in order to address the credit assignment problem in MARL through the world model learning process?
>
> **Response**: Thanks for the constructive comment. As shown in response to **Q1**, we suppose currently the learning w.r.t. the reward prediction in our algorithm is implicitly a simple credit assignment. In terms of learning a different reward for each agent, we have not yet considered using different individual reward labels. But if the current team reward can be decomposed into different reward labels according to the actual individual contribution, it is intuitively promising for further enhancing policy learning in imaginations of the world model. Generally speaking, credit assignment can be approached through non-learning-based methods (e.g., manually assigning rewards based on each agent's specific actions) or learning-based methods (e.g., counterfactual inference via the value function as in COMA[2]). Non-learning-based approaches tend to introduce excessive human priors, making the learning world model overly domain-specific and lacking generalization capability. On the other hand, jointly training learnable credit assignment and the world model may face the challenge from co-training. We would leave this to the future work.
>
> **Q3**: In the training process of the world model, how are the trajectories used as labels obtained? Also, please discuss what should be done in complex scenarios when there is no good initial policy to generate the trajectories.
>
> **Response**: Thanks for the question. The trajectories for world model training is collected with a policy which is randomly initialized and iteratively trained inside the learned world model. The learning pipeline is stated in Line 164-166. In complex scenarios, we can still deploy the same procedure to improve the sample efficiency of policy learning.
>
> **Q4**: Please explain in detail the role of learning $\gamma$ in the overall method, and why it cannot be replaced with a constant.
>
> **Response**: Thanks for the question. Regarding learning discount $\gamma$ prediction, the labels $\gamma = 0.99$ for time steps within an episode and are set to zero for terminal time steps. Given an imagination horizon $H$, the predicted discount sequence $\hat{\gamma}_{1:H}^{i}$ is used to weigh the loss terms of the actor and critic of agent $i$ by the cumulative predicted discount factors to **softly account for the possibility of episode ends** while computing the cumulative return of the imaginations.

---

> > ### Author Response · Authors · 2024-11-22
> > **Response to Reviewer GJdy (Part 2/2)**
> >
> > **Q5**: How is the codebook $\mathcal{Z}$ initialized, and does the initialization affect the learning outcomes under different conditions?
> >
> > **Response**: Thanks for the question. In our algorithm, we adopt a open-source VQ-VAE implementation mentioned in Line 809, which uses *kaiming_init* to initialize the codebook $\mathcal{Z}$. As for the initialization effect on the learning outcomes, we tried orthogonal initialization for the codebook in the early stage of the experiment and did not observe any remarkable performance differences compared with the one with the default initialization (i.e., *kaiming_init*).
> >
> > **Reference**
> >
> > [1] Hafner, Danijar, et al. "Dream to control: Learning behaviors by latent imagination." arXiv preprint arXiv:1912.01603 (2019).
> >
> > [2] Foerster, Jakob, et al. "Counterfactual multi-agent policy gradients." Proceedings of the AAAI conference on artificial intelligence. Vol. 32. No. 1. 2018.

---

### Author Response · Authors · 2024-12-01
**General Response**

Dear reviewers and AC,

We sincerely appreciate your valuable time and effort spent reviewing our manuscript.

We propose MARIE, a sample-efficient model-based MARL algorithm driven by an effective Transformer-based multi-agent world model, which is built by casting the decentralized agent dynamics into sequence modeling over discrete tokens to solve the *scalability* issue, and incorporating it with a centralized agent-wise aggregation to solve the *non-stationarity* issue. As reviewers highlighted:
1. Our method, especially the integration of decentralized dynamics and centralized aggregation, is a notable strength, relatively straightforward and well-motivated (all);
2. Our method is demonstrated by extensive experiments and analysis (GJdy, qbLx, Cbz9);
3. The use of the Perceiver Transformer for centralized aggregation is innovative and provides valuable insights (ysTM, Cbz9).

We appreciate your constructive feedback on our manuscript. In response to the comments,we have carefully revised and enhanced the manuscript as follows:
1. Additional re-implementation, adaptation and evaluation of MAMBA evaluation of MAMBA on MAMujoco (Appendix E.1);
2. Additional experiments on SMACv2 (Appendix E.2);
3. Comparison with existing Transformer-based world models (Appendix E.4);
4. Comparison against MAMBA with different imagination horizon (Appendix E.5);
5. Additional Discussion on the related works reviewers provided (Appendix F);
6. Rigorous statistical evaluation by using RLiable (Appendix G);
7. Limitation discussion on using limited seeds (Section 5);
8. Increased font size of figures in the main text to enhance readability;
9. A follow-up conclusion in the caption Figure 3~6 to improve the presentation.

Besides, we have also provided the results of essential experiments including:
1. The comparison on computational cost and performance between different means of aggregation (Response to Reviewer ysTM);
2. The evaluation of the accuracy of MARIE's discount prediction under different imagination horizon (Response to Reviewer ysTM).

The above results would be also incorporated into the future revision right after having access to revising our manuscript. Lastly, in the current revised manuscript, these updates are temporarily highlighted in blue for your convenience to check. We sincerely believe that these updates could help us better deliver the benefits of the proposed MARIE to the multi-agent community.

Thank you very much,

Authors.

---

### Meta-Review · Area_Chair_QdDd · 2024-12-19

**Metareview:**

The paper introduces MARIE, a Transformer-based architecture designed to improve sample efficiency by enhancing the accuracy of multi-agent world modeling. The authors aim to address challenges of world modeling in multi-agent reinforcement learning (MARL), particularly the scalability and non-stationarity issues, by using decentralized local dynamics combined with centralized aggregation through a Perceiver Transformer. Reviewers believe that the method proposed by the authors is somewhat novel, the writing is good, and the experimental setup is very clear. Considering that none of the reviewers expressed clear support for the paper (scores were 6665) and were not positive about the rebuttal, AC encourages the authors to supplement the experiments based on the reviewers' opinions and carefully revise the paper, believing that its quality can be greatly improved.

**Additional Comments On Reviewer Discussion:**

The main point of contention regarding this paper is the sufficiency and rigor of the experiments, such as the comparison setup, hyperparameter selection, and code-level details. Although the authors made significant efforts during the rebuttal period, the reviewers clearly pointed out that these concerns were not truly resolved. None of the reviewers expressed clear support for the paper (scores were 6665) and were not positive about the rebuttal.

---

### Decision · Program_Chairs · 2025-01-22

Reject